# Study of shear-plastic slip mechanism based on TC4 titanium alloy

**Bo Hu** **[1], Zichuan Zou[2], Pengfei Tian[3], Nian Xiao[4], Sen Yuan[1]\*, Xianfeng Zhao[3]\***

**1** School of Mechanical Engineering, Guizhou Institute of Technology, Guiyang, China, **2** School of Mechanical & Electrical Engineering, Guizhou Normal University, Guiyang, China, **3** School of Mechanical Engineering, Guizhou University, Guiyang, China, **4** Shizhen College of Guizhou University of Traditional Chinese Medicine, Guiyang, China

\* syuan@git.edu.cn; xfzhao1@gzu.edu.cn

## Abstract

The stagnation point and dead metal zone in the cutting process directly or indirectly affect the chip formation and stress distribution, while the stress distribution in the machining process determines the plastic slip direction of the material. Aiming at the current insufficient research on the dead metal zone and stagnation point theory, this paper divides the cutting process into rounded edge contact stage and rounded edge-rake face contact, constructs a slip line field model with dead metal zone based on the stress distribution and pressure distribution of the two stages, calculates the slip line field through the Cauchy problem, and plots the slip line field through the secondary development port in SOLIDWORKS. The dead metal zone model is based on the stress distribution of the obtuse circular contact, and the stagnation point occurs at the critical condition of the elastic-plastic transition of the material, i.e., at the maximum shear stress of the process. The dead metal zone and stagnation point are examined based on simulation, and the slip line field model is verified experimentally. The results show that the dead metal zone model can be predicted more accurately when the tool rake angle is 15° or less, and the greatest influence on the stagnation point is the tool rake angle and the radius of the rounded edge of the tooltip, and the slip line field model containing the dead metal zone can more accurately reflect the plastic slip of the real cutting process. It can be seen that the dead metal zone model, stagnation point model, and slip line field model illustrate the cutting mechanism of the elastic and plastic phases of the cutting process, which lays a research foundation for the subsequent study of tool wear, chip formation, and machining surface quality.

## 1. Introduction

The concept of slip line field was first proposed by Hencky H. [1] in 1923, and was later introduced and widely used in various fields including mechanical, civil, and

**Data availability statement:** All relevant data are within the paper and its Supporting information files.

**Funding:** The Guizhou Provincial Youth Science and Technology Talents Growth Project(Grant No.QJJ[2024]163); Higher Education Engineering Research Center of Guizhou Province (Grant No. QJJ[2023]040); Guizhou Association for Science and Technology New Quality Qianyan Leading Project - Youth Voyage Program 2025 (XZQYXM-01-10).

**Competing interests:** The authors have declared that no competing interests exist.

geological. It was introduced to mechanical engineering to explain the plastic forming of metals, and then gradually used to solve the cutting mechanism problems such as cutting force, size effect chip forming, etc. In the 1950s, Merchant M. E. [2] first established a single slip line field model (see Fig 1a) in his study of the plastic forming mechanism of orthogonal cutting, and then Lee E. H. et al [3] built based on the single slip line field established by him. Established a single slip line field based on the slip line field distribution based on the plastic deformation zone of the front tool face (deformation zone II), and the shear angle formula obtained from this model became a generalized formula, and the model also considered the existence of the dead zone, which was verified by Palmer W. B. et al [4] in a subsequent study. However, the model does not consider the changes in the third deformation zone, and with the increasing requirements for cutting accuracy in recent years, the third deformation zone and the plowing force acting in it become more and more important. Subsequently, many new slip line models have been established based on this model, which is mainly used to study the prediction of plowing force, the determination of dead zones, diversion points chip formation mechanism [5], and so on.

After decades of exploration, the research on the positive leading angle slip line model has been developed rapidly to some extent, among which the model established by Waldorf D. J. et al [6] in 1998 is the most classical, as shown in Fig 1b, in the study of the significance of the plowing effect in the finishing process, a positive leading angle slip line model for predicting the plowing force was established and verified by experiments, which represents the relationship between the plowing force and the relationship between plowing force and tool edge radius. However, this model is only applicable to large front angles and not applicable to small front angles, so this model is often improved by later scholars and used in the study of cutting mechanisms, for example, Rebaioli L. et al [7] referred to the Waldorf model and applied it to the micro-machining process and predicted the cutting force under different feeds and cutting speeds, and also proposed an experimental method on the slip line field model; Sun Q. Q. et al [8] proposed a new method to predict the cutting forces during micro end milling, and Waldorf's slip line solution was utilized to determine the plowing force coefficient in the prediction process; Uysal A. et al [9] established a slip line model for worn tools considering the presence of dead zones based on Waldorf's model, which was used to study the plowing and friction forces generated by tooth wear, chip upward and downward radii, chip thickness, and the cutting force. Radius, chip thickness, main shear zone thickness, and dead zone area Fig 2.

Compared to the common positive rake angle cutting tools, the negative rake angle cutting machining process increases the strength of the tool, improves the heat transfer from the cutting edge to the shank, and increases the tool life, because of these properties, cutting tools with negative rake angle are widely used [10]. With a large number of researchers on negative rake angle cutting, many researchers have established a negative rake angle slip line field model and also observed the existence of dead zones, the appearance of dead zones determines the shunt point becomes inconsistent, so the slip field model is also very different, it can be

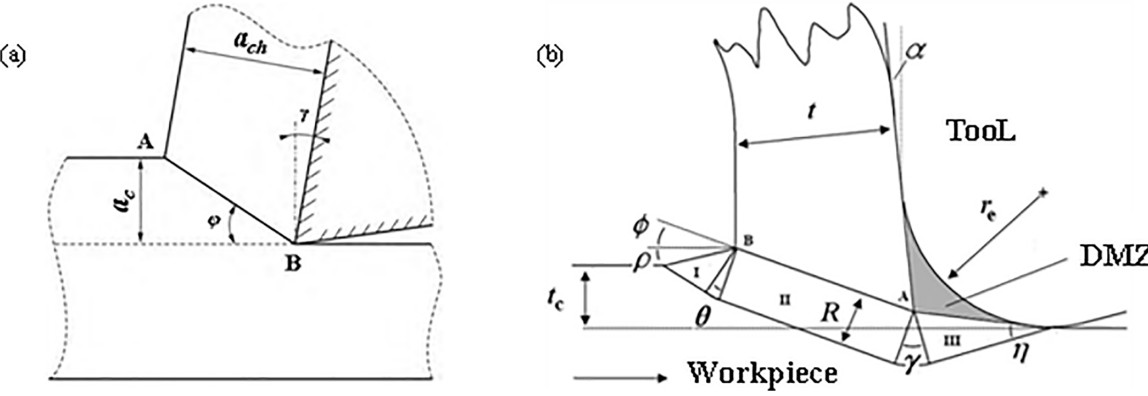

**Fig 1. Shear surface modeling by Merchant M. E. [2,6].**

said that the establishment of slip line field can not be bypassed by dead zones, and the dead zones are also differently understood.

Compared to the study of positive rake angle slip line field, the study of negative rake angle slip line field is relatively late. In 1981, Abebe M. [11] was the first one to propose a dead zone model for machining with negative rake angle tools, and based on this, he established a slip line field model and elaborated the basic concepts of negative rake angle slip line field model. In the following year, Kita Y. et al [12], in order to observe the stagnation behavior of the material on the front face of the tool, established a negative fore-angle slip line model based on the Abebe model, and at the same time investigated the effect of cutting speed on the shape of the dead zone of the material generated in front of the tool's front face, and came to the following conclusions: the initial position of the stagnation tip not only determines the chip size, but also determines whether or not a chip will be formed; the material's grinding process movement is determined by the mixing mode of metal flow and depends on the position of the stagnant tip, the friction acting on the tool surface and the proportion of metal flowing upward and downward; in 1987, Kita Y. et al [13] used a negative rake angle slip line model to observe the flow mechanism of the material in front of the cutting edge in front of the tool on the basis of the literature [12] and elaborated on the relationship between the rake angle and the shear angle to get the same as that in literature [2] Conclusion; then Petrky H. [14] proposed a new negative front angle slip line field model, which elaborated the metal flow direction and dead zone changed according to the value of τ (shear stress)/k (material shear flow stress) and negative inclination angle.

Most of the negative front angle slip line field models established in the previous period elaborate on the plastic flow mechanism at the front tool face, and the proposed theoretical values have some errors with the actual values. With the continuous improvement of the models, many researchers began to study the effects of cutting force and dead zone based on the negative slip line field models established by the previous authors. In 2005, Fang [15], based on the model of Lee and Shaffer [3], proposed a negative rake angle slip line model, based on which he investigated how negative rake angle tools and cutting speeds affect tool-chip friction, and how tool-chip friction further affects machining performance, such as the ratio of cutting force to thrust, the chip thickness ratio, the geometry of the shear zone, and the geometry of the stagnant zone of the material flow near the front face of the tool, as well as analyzing and comparing the different effects on chip friction between the positive rake angle and the negative rake angle.

In 2012, Ozturk S. et al [16] proposed a new sliding line model to be used for simulating the orthogonal cutting process and its associated trajectories for blunt circular tools. The model consists of eight regions that involve dead zones. In the mathematical expression of the new model, the matrix technique of Dewhurst and Collins for numerically solving the slip line problem is used.

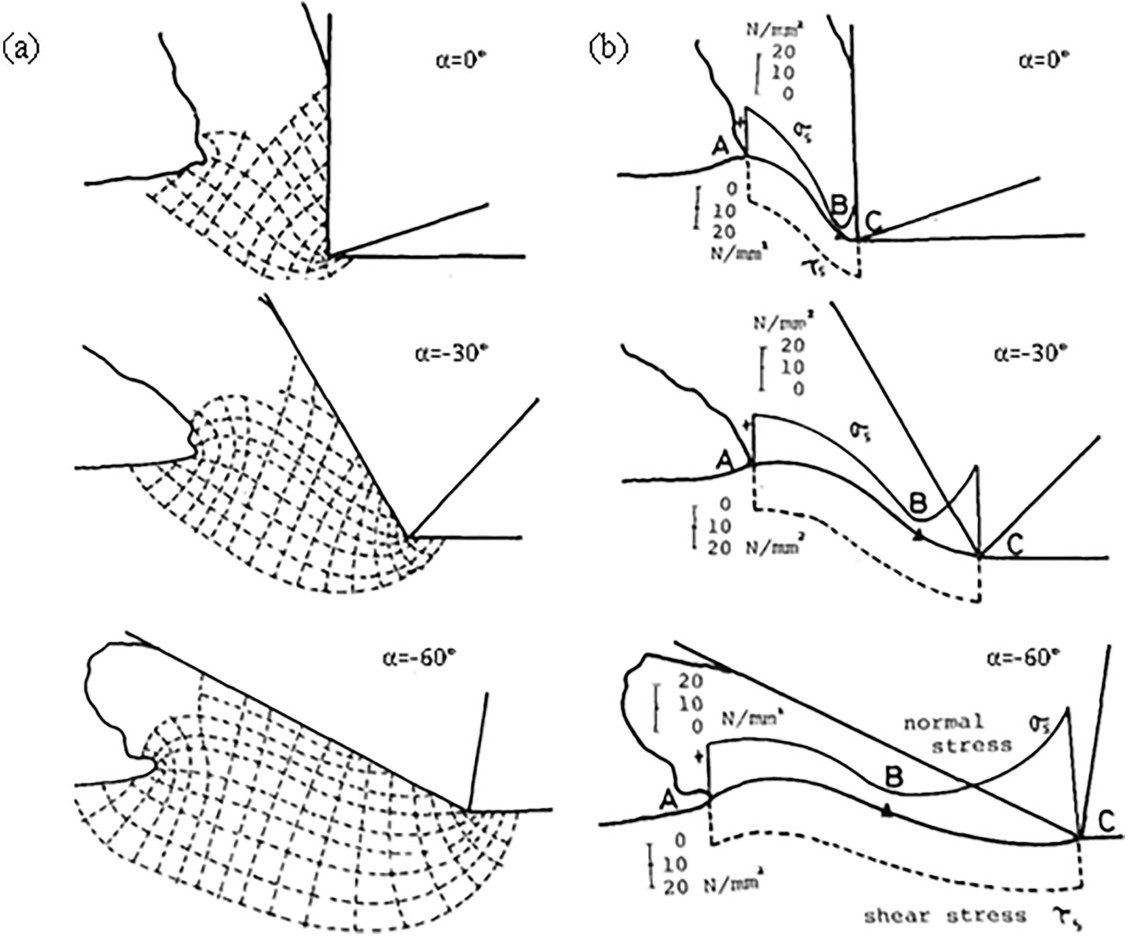

**Fig 2. Kita Y. Slip line field [12].**

In metal extrusion processing, there is a special area known as the dead zone, which is located in the tip of the cutter, and the material cut at the junction of the dead zone will be caused by the metal in the dead zone with the cutting feed will gradually with the plastic deformation area of the metal flows out of the tip of the cutter, resulting in the volume of the dead zone is reduced. However, in the non-lubricated cutting forward extrusion, the dead zone plays a protective role in preventing the processing of surface defects, oxides, and debris into the plastic deformation zone and affects the surface quality of the product. However, if lubricated cutting is used, oxidation is serious, and the protective effect of the dead zone is weakened, which may allow oxides and impurities to enter the product surface and affect product quality. The dead zone and shunt point are concepts that have been gradually unified in the plastic deformation process of cutting, and a large number of scholars have verified them through experiments and simulations. In recent years, with the develop-ment of blunt round tools, due to their ability to provide good cutting toughness, and at the same time reduce the wear of the tool, making the extension of its service life, but also lead to the formation of metal dead zone (DMZ). Since the phe-nomenon of material buildup during cutting was first described scientifically in 1890, a lot of theoretical studies have been carried out. In 1967, Usui E. et al [17], in establishing the slip line field, suggested that the conditions for the existence of the dead zone and the mechanical behavior of the cutting tool's front face are still not clear, especially when the metal cut-ting is performed at low cutting speeds or when the cutting tool is cutting with a large negative rake angle, it is possible to

observe a very large plastic zone; Hirao M. et al [18] investigated the accumulation of material on the front tool face during the cutting process and concluded that the dead zone has a direct effect on the roughness of the cutting surface, while Palmer W. B. et al [19] concluded that the direction of material curl is dependent on the negative tool inclination angle and that the geometry of the cutting tool is an important factor in the morphology and generation of the dead zone. In the above model, the metal dead zone is usually considered to be stagnant and immobile for the front face of the tool, and therefore, the material flow is closely related to the dead zone geometry. Until 2002 it was concluded through experimental and analytical studies by Shi G. et al [20] that the dead zone is mainly dependent on the geometry of the chamfered portion rather than the cutting conditions.

With the in-depth study of the cutting mechanism and experimental conditions, the view that the divergence point is located on the blunt circle boundary of the tool has changed, while some scholars have also elaborated on the effect of the cutting process parameters on the dead zone and divergence point. In 1971 Komanduri R. [21] proposed that in the metal cutting process, the flow of the material over the cutter face is bi-directional, with one part of the material flowing under the cutter (friction or plowing) and the other part of the material flows from the chip up the face of the tool with a stationary point which depends on the size of the rake angle; Weule H. et al [22] based on the theoretical study of microfinishing cutting suggested that lower cutting speed is an important factor in the formation of stagnant zones, and poor surface roughness was also observed; Kountanya R. K. et al [23] found out that the use of round edged tool to cut brass material orthogonally, stagnation zone is formed in front of the cutting edge and it disappears when the round edged tool cuts zinc material orthogonally; the study of Ozturk S. et al [16] in 2012 also showed that in orthogonal cutting model when the value of the negative rake angle decreases, the stagnation zone will increase and the ratio of the pressure to the cutting force will increase, which will result in improved quality of the machined surface.

| Nomenclature | |
|---|---|
| $q_0$ | Maximum distribution force |
| $\gamma$ | Rake angle |
| $\alpha$ | Mold angle |
| La | Rake face contact length |
| $E^*$ | Elastic modulus equivalent |
| Lb | Contact length of arc segments |
| R | Rounded edge |
| $\alpha_1$ | Angle at chord length for blunt circle contact |
| $\sigma_x$ | X-axis stress in a plane coordinate system |
| $\sigma_m$ | Hydrostatic pressure |
| $\sigma_y$ | Y-axis stress in a plane coordinate system |
| v | Poisson's ratio |
| $\tau_{xy}$ | Shear stress in the xy plane in a planar coordinate system |
| r | Contact half-width |
| d | Maximum distance between dead zone arc and rounded edge |
| $\sigma_1$ | First principal stress |
| $\theta$ | Rotation angle in polar coordinates |
| $\sigma_2$ | Second principal stress |
| $\sigma_s$ | Yielding strength |
| $\sigma_3$ | Third principal stress |
| A | Load factor |

With the continuous innovation of finite element, the validation for dead zone is also found in cutting simulation, Wan L. et al [24] numerically investigated the friction in cutting process using ABAQUS software and proposed a finite element model based on the Arbitrary Lagrangian-Eulerian (ALE) method for simulation of the cutting process and to study the

effect of friction on the dead zone; Movahhedy M. R. et al [25] performed several thermal Arbitrary Lagrangian-Eulerian (ALE) simulations of chip formation during cutting of different types of chamfering tools and observed the phenomenon of DMZ formation in finite element simulations and concluded that the chamfer angle does not have a significant effect on chip formation due to the presence of DMZ; Karpat Yiğit et al [26] proposed a new analytical model of DMZ, which exists below the chamfered edge or near the worn part under different cutting conditions and mainly depends on the geometric profile of the tool edge rather than the cutting variables. Wan L. et al [27] investigated the effect of tool edge geometry and friction coefficient on DMZ formation through finite element simulation and experimental validation and found that the tool edge geometry and friction coefficient are both responsible for DMZ formation.

In general, the study of the slip line of the cutting process can effectively understand the flow and deformation of the material, which is an important way to solve the cutting mechanism problems such as cutting force, size effect, and chip. In addition, the slip line theory provides a new calculation method, which can reasonably analyze and calculate the cutting force and accurately predict and optimize the cutting parameters, thus improving the machining quality, reducing tool wear, and improving tool life. The formation mechanism of dead zone and shunt point has yet to be systematically explored, the generation of dead zone and shunt point is the main factor directly affecting the machining surface quality, the study of the mechanism of dead zone and shunt point can effectively reduce the stagnation and separation of the material in the cutting process, to improve the tool service life and the machining effect, and to improve the production efficiency and product quality.

## 2. Establishment of theoretical model

### 2.1. Slip line field establishment

**2.1.1. Rounded edge sliding line field.** After the rounded edge of the tool and the workpiece contact, due to the surface of the workpiece and the rounded edge of the outer surface of the fit, the contact surface of the material is considered to be circular, to establish the contact surface of the model for a section of the arc, when the rounded edge of the tool and the material contact as shown in Fig 3a, in the contact area, the pressure distribution function is:

$$q(y) = q_0 \sqrt{1 - \left(\frac{y}{b}\right)^2} \tag{1}$$

where: $q_0 = \sqrt{\frac{FE}{\pi L(1-v^2)r}}$

The blunt circular contact stress distribution function is [28]:

$$\begin{cases} \sigma_x = -A(2(\theta_2 - \theta_1) + (\sin 2\theta_2 - \sin 2\theta_1)) \\ \sigma_y = -A(2(\theta_2 - \theta_1) - (\sin 2\theta_2 - \sin 2\theta_1)) \\ \tau_{xy} = A(\cos 2\theta_2 - \cos 2\theta_1) \end{cases} \tag{2}$$

where: $A = \frac{q_0}{2\pi}$

At its contact surface, the critical condition for stress distribution is:

$$\sigma_x = \sigma_y = q(y) \tag{3}$$

$$\theta = \arcsin \frac{y}{r} \tag{4}$$

From the yield condition $(\sigma_x - \sigma_y)^2 + 4\tau_{xy}^2 = 4k^2$:

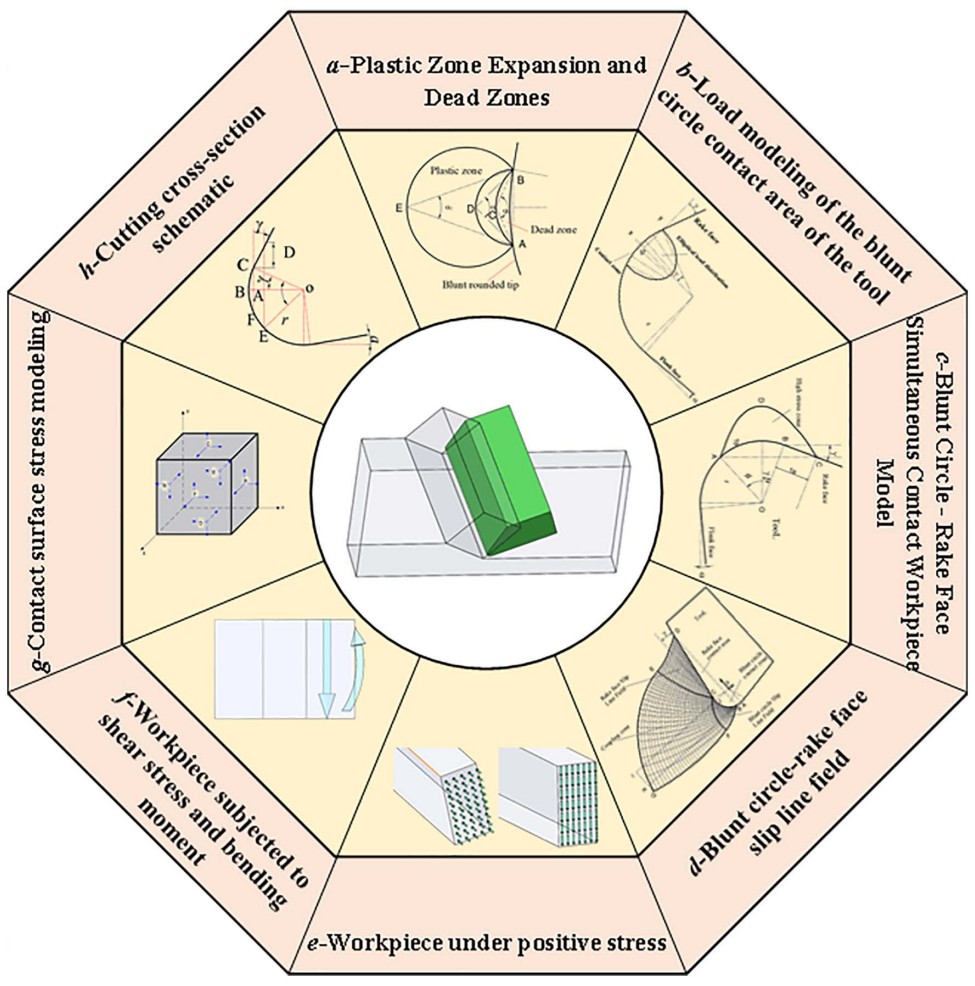

**Fig 3. Large negative leading angle slip line modeled by Fang [15].**

$$2A \sin \alpha = \frac{q_0}{\pi} \sin \alpha = k \tag{5}$$

which: $\begin{cases} k = \frac{\sigma_s}{2} \\ k = \frac{\sigma_s}{\sqrt{3}} \end{cases}$

  Therefore, it can be known that the material yields when $\frac{q_0}{\pi} \sin \alpha > k$, and the specific parameters of the obtuse circular contact are shown in Fig 3h. Where α is the angle corresponding to the obtuse circular contact width for the chord, and kπ is a constant value. When $\alpha = \frac{\pi}{2}$ $q_0$ takes the smallest value, plastic deformation is most likely to occur on the circle with a contact width of diameter, and then expand to both sides, there is a region in the middle of the contact surface and the plastic zone is a dead metal zone, as shown in Fig 3b.

  **2.1.2. Flank face slip line field.** The cutting process can be viewed as a result of the tool's positive stress, shear stress, and bending moment on the workpiece material (as shown in Fig 3e, 3f), with the workpiece feeding action, the rake face and the rounded edge contact the workpiece material at the same time, in front of the tool to form a high-stress region, the form of the contact and the high-stress region as shown in Fig 3c.

In [Fig 1c](): the radius of the rounded edge is r, the contact length of the rake face of the tool is La, the center angle of the rounded edge contact arc is φ, and the length is Lb. A high-stress area is formed in front of the contact area, and the high point of the high-stress area is at point D. Therefore, the contact length of the rake face of the tool is La.

Therefore the contact length of the rake face of the tool is

$$L_a = \frac{4P}{\pi E \sin\gamma} - r$$

(6)

Pressure distribution on the rake face of the tool is derived based on Hertzian contact mechanics.

$$p(y) = \frac{(P + \frac{2M}{b^2}y)}{\pi\sqrt{b^2 - y^2}} = \frac{P}{\pi a}\frac{1 + \frac{y}{b}}{\sqrt{1 - \left(\frac{y}{b}\right)^2}}$$

(7)

The resulting theoretical slip line field is shown in [Fig 3d]().

In [Fig 3d](), the center of the Rounded Edgeis point O, the radius of the Rounded Edgeis r, points A, B, J, and C are located on the rounded edge, point A is the lowest point of the tool, point B is the lowest point of the Rounded Edgein contact with the workpiece material, point J is the intersection of the horizontal line and the rounded edge, point C is the point of incision of the Rounded Edgeand the straight line segment of the front cutter surface, and point BJC is the line of contact of the Rounded Edge with the workpiece material. Cd is the line of contact of the front cutter surface with the workpiece material. The slip line field formed by the rounded edge of the tool is the BJCF triangular area, and the slip line field formed by the rake face of the tool is the CDE triangular area.

FCEG is the coupling area formed by the two triangular slip line fields, and point G is the farthest point of the slip line field. When the tool starts to contact the workpiece material, the farthest point of the slip line field is located above the lowest point of the tool, and with the continuous feeding of the tool, the farthest point of the slip line field formed continues to expand forward and downward, and the area of the slip zone is increasing.

In the slip zone, the rounded edge slip line field formed in the region is a high-pressure area, as long as the tool's rounded edge contact position is determined, this high-pressure area is determined. The two boundary lines CF and BF of the rounded edge slip line field.

The entire triangular slip line field is divided into three regions: the rounded edge slip line field, the rake face slip line field, and the coupling region. The rounded edge slip line field and the rake face slip line field form a stagnant layer (cold weld zone or dead metal zone) due to the high-pressure zone created by the tool compressing the material. The entire slip line field, at the right speed and temperature, will form a the built-up edge. Shedding of the coma typically occurs at about half the thickness of the coma. The direction of shedding may be downward or upward, depending on whether the stagnation point is located above or below the lowest point of the tool's rounded edge.

## 2.2. Slip line calculation

By the equation of the slip line in the plastic phase.

$$\begin{cases} \sigma_m - 2k\theta = \xi & \alpha \\ \sigma_m + 2k\theta = \eta & \beta \end{cases}$$

(8)

Speed coordination [equation (9)]():

$$\begin{cases} d\dot{u} - \dot{u}\,d\theta = 0 & \alpha \\ \phantom{d}_\alpha \phantom{_\beta} \\ d\dot{u} - \dot{u}\,\alpha d\theta = 0 & \beta \\ \phantom{d}_\beta \end{cases}$$

(9)

If the slip line field and the values of $\xi$ and $\eta$ are known, $\sigma_m$ and $\theta$ can be found for the points within the coordinates. And $\sigma_x$, $\sigma_y$, and $\tau_{xy}$ for the contact process can be found according to Eqs. (10), (11) and (12)

$$\sigma_x = -\frac{2P\cos^4\theta}{\pi a} \tag{10}$$

$$\sigma_y = -\frac{2P\sin^2\theta\cos^2\theta}{\pi a} \tag{11}$$

$$\tau_{xy} = -\frac{2P\sin\theta\cos^3\theta}{\pi a} \tag{12}$$

where P is the concentrated force, a is the feed depth, r is the distance from the point to the center of the circle and $\theta$ is the angle between the origin to r and the x-axis. The stresses on the workpiece are shown in Fig 1g.

The establishment of the slip line field is divided into three types of problems, depending on the $\sigma_x$, $\sigma_y$, and $\tau_{xy}$ out of Eqs. (10), (11) and (12) and the boundary conditions:

Type 1 Cauchy problem. In Fig 4a, a line segment in the right-angle coordinate xoy plane is assigned to $\sigma_m$ and $\theta$ with the stipulation that all lines intersect one time and only once, and are everywhere not tangent to the slip line. Thus, based on the Cauchy problem, it follows that either side of that line segment establishes a unique slip line field within the curved triangle ABC with the line segment as the base.

Type 2 Riemann problem. In Fig 4b, given functions $\sigma_m$ and $\theta$ on the slip line segments OA and OB, a unique slip line field is established within the curved quadrilateral OACB with these two slip line segments as neighbors. Special case I of this problem is where one of the two slip lines recedes to a point, thus creating a sector field, as shown in Fig 4c.

Type 3 Hybrid Problems. A mixture of the Cauchy problem and the Riemann problem.

In this paper, the Cauchy problem such as Fig 1a is used as an example to illustrate the numerical calculation method of the slip line field. Given $\sigma_m$ and $\theta$ on AB, the AB segment can be divided into finite small segments, and two characteristic lines are drawn from each sub-point, so the ABC area is divided into finite grids, and the intersection points of the grids are denoted as (1,1), (2,2), etc., as shown in Fig 1a. Based on Hankey's equation:

$$\begin{aligned} \alpha: \quad & \sigma_m - 2k\theta = \xi \\ \beta: \quad & \sigma_m + 2k\theta = \eta \end{aligned} \tag{13}$$

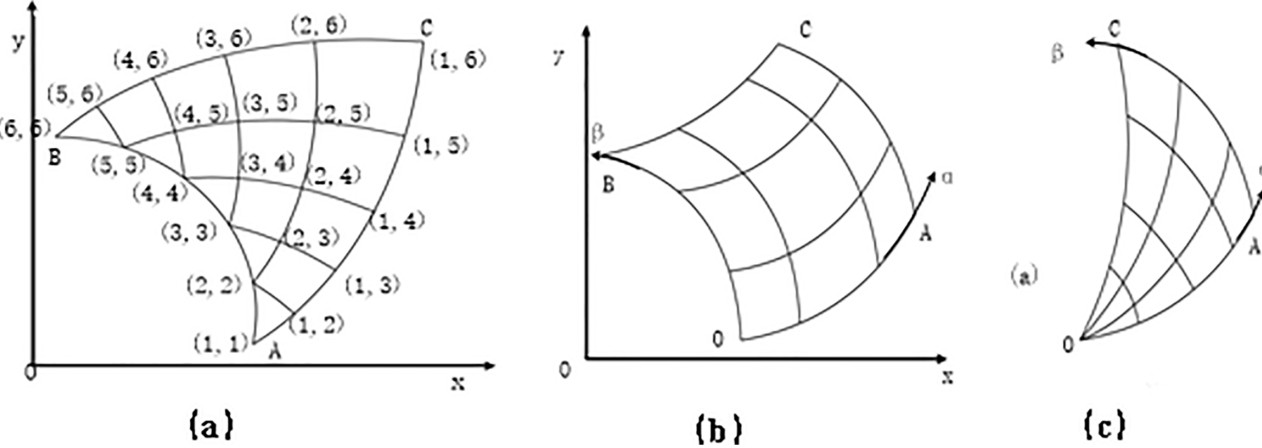

**Fig 4. Slip line modeled by Ozturk S [16].**

This can be extrapolated to arrive at

$$\begin{array}{ll} \alpha: & \sigma_{m_{11}} - 2k\theta_{11} = \sigma_{m_{21}} - 2k\theta_{21} \quad (a) \\ \beta: & \sigma_{m_{22}} + 2k\theta_{22} = \sigma_{m_{21}} + 2k\theta_{21} \quad (b) \end{array} \tag{14}$$

Adding up (a) and (b) in equation (14):

$$\sigma_{21} = \frac{1}{2}(\sigma_{22} + \sigma_{11}) + k(\theta_{22} - \theta_{11}) \tag{15}$$

Subtracting (a) and (b) in equation (14):

$$\theta_{21} = \frac{1}{4k}(\sigma_{22} - \sigma_{11}) + \frac{1}{2}(\theta_{22} + \theta_{11}) \tag{16}$$

Generalizing Eq. (15) and Eq. (16) to any point yields:

$$\begin{array}{l} \sigma_{m,m-1} = \frac{1}{2}(\sigma_{m,m} + \sigma_{m-1,m-1}) + k(\theta_{m,m} - \theta_{m-1,m-1}) \\ \theta_{m,m-1} = \frac{1}{4k}(\sigma_{m,m} - \sigma_{m-1,m-1}) + \frac{1}{2}(\theta_{m,m} + \theta_{m-1,m-1}) \end{array} \tag{17}$$

Eq. (17) is the established slip line field generalized coordinates, bringing in the relevant parameters that can be found in the given region $\sigma_m, \theta$

By mapping to the instructional coordinate system, this can be computed as follows:

$$\begin{array}{ll} \alpha: & \frac{\Delta y}{\Delta x} = \tan\theta \\ \beta: & \frac{\Delta y}{\Delta x} = -\cot\theta \end{array} \tag{18}$$

Establish the equation as:

$$\begin{array}{ll} y_{m,n} - y_{m-1,n} = (x_{m,n} - x_{m-1,n})\tan\frac{\theta_{m,n} + \theta_{m-1,n}}{2} & (a) \\ y_{m,n} - y_{m,n+1} = -(x_{m,n} - x_{m,n+1})\cot\frac{\theta_{m,n} + \theta_{m,n+1}}{2} & (b) \end{array} \tag{19}$$

Subtracting (a) and (b) from equation (17):

$$x_{m,n} = \frac{y_{m,n+1} - y_{m-1,n} + x_{m,n+1}\cot\frac{\theta_{m,n} + \theta_{m,n+1}}{2} + x_{m-1,n}\tan\frac{\theta_{m,n} + \theta_{m-1,n}}{2}}{\left(\tan\frac{\theta_{m,n} + \theta_{m-1,n}}{2} + \cot\frac{\theta_{m,n} + \theta_{m,n+1}}{2}\right)} \tag{20}$$

$$y_{m,n} = (x_{m,n} - x_{m-1,n})\tan\frac{\theta_{m,n} + \theta_{m-1,n}}{2} + y_{m-1,n} \tag{21}$$

## 2.3. Stagnation points and dead metal zones

In the study of the slip line field, a large number of scholars have explored the stagnation point and dead metal zone in the cutting process, so the stagnation point and dead metal zone have been an important means to study the cutting mechanism. In general, a large number of scholars have studied that the dead metal zone and stagnation point obtained by different theories and experiments under the same cutting parameters are not the same, in addition, the researchers have also concluded that the size of the dead metal zone changes with the change of cutting parameters and geometrical parameters, and the stagnation point changes with the change of the size of the dead metal zone.

 

By analyzing the stresses in the obtuse circular contact phase, the three-way stress expansion for this contact process can be expanded in polar coordinates as follows

$$\begin{cases} \sigma_1 = -2A\left(\alpha + \sin\alpha\right) \\ \sigma_2 = -2A\alpha \\ \sigma_3 = -2A\left(\alpha - \sin\alpha\right) \end{cases} \tag{22}$$

Where: A is the load factor and $A = \frac{q_0}{2\pi}$, $\alpha = \theta_2 - \theta_1$

According to the maximum tensile stress criterion $\sigma_1 \geq \sigma_b$ when the material occurs damaged. By the maximum elongation line, strain theory damages critical conditions $\varepsilon_1 \geq \varepsilon_\mu$ when the stress critical conditions can be obtained as follows

$$\sigma_1 - \mu(\sigma_2 + \sigma_3) \geq \sigma_b \tag{23}$$

Bringing Eq. (22) into Eq. (23) yields

$$2A\alpha(2\mu - 1) - 2A(1 + \mu)\sin\alpha \geq \sigma_b \tag{24}$$

Establishing the function $f(\alpha) = 2A\alpha(2\mu - 1) - 2A(1 + \mu)\sin\alpha$ yields

$$\begin{cases} f'(\alpha) = 2A(2\mu - 1) - 2A(1 + \mu)\cos\alpha \\ f''(\alpha) = 2A(1 + \mu)\sin\alpha \end{cases} \tag{25}$$

The differential equation can be solved by making $f'(\alpha) = 0$

$$\cos\alpha = \frac{2\mu - 1}{1 + \mu} \tag{26}$$

The limit angle is obtained as

$$\alpha = \cos^{-1}\frac{2\mu - 1}{1 + \mu} \tag{27}$$

By reviewing the literature [2] to get the titanium alloy TC4 Poisson's ratio $\mu = 0.3$, which will be brought into the formula (27) $\alpha_0 = 107.92°$. And because $f''(\alpha) = 2A(1 + \mu)\sin\alpha = 2.46A > 0$, when $\alpha_0 = 107.92°$ to get the minimum value of 3.918 A. To sum up, with the continuous feeding of the tool, the function $f(\alpha)$ to reach the stress boundary conditions when the material is damaged, through the Matlab plotting of the function $f(\alpha)$ curves and the Rounded Edgeschematic shown in Fig 5.

As can be seen in Fig 6, the destruction of the material occurs first $\alpha_0 = 107.92°$, which corresponds to the rounded edge model that is on the iso contact stress curve $\alpha = 107.92°$, which is manifested as a circular arc with the contact length as the chord. However, with continuous feeding, its contact width will increase, so its damage area will also change, see Fig 7.

As can be seen in Fig 8, with the feed of the tool its limit damage range is the annular region, when the cutting conditions to reach a steady state can be determined, and its boundary outside the region that is the critical region of the material crack unfolding, within the boundary region that is the formation of the dead metal zone. The slip-type cracks formed under this boundary condition will form open cracks on the boundary with the feed, from which the arc radius of the boundary outside the dead metal zone can be calculated as

$$r = \frac{a}{\sin\alpha_1} \tag{28}$$

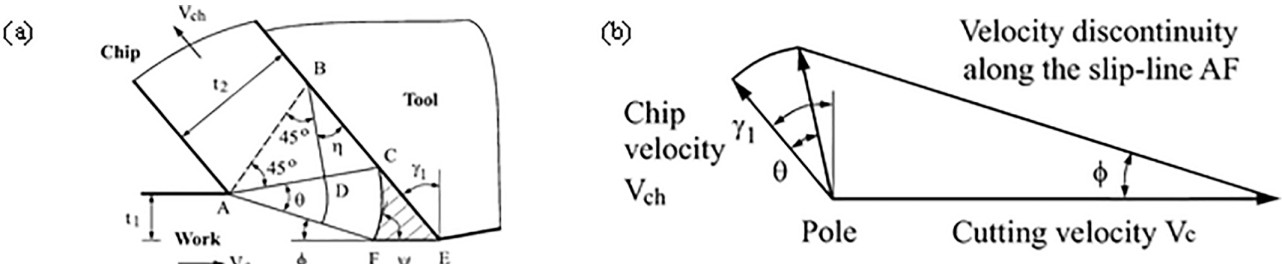

**Fig 5. Slip line field modeling of rounded edge contact and rounded edge-rake face contact.**

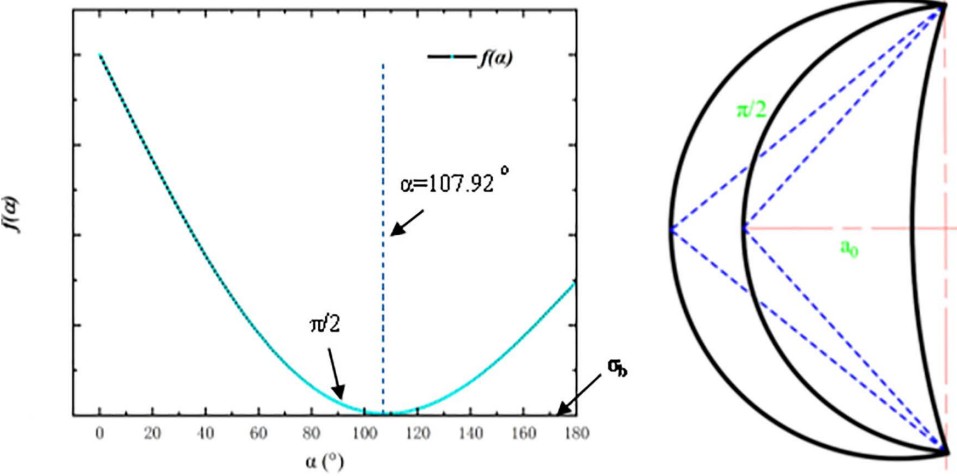

**Fig 6. Schematic diagram of the Cauchy and Riemann problems.**

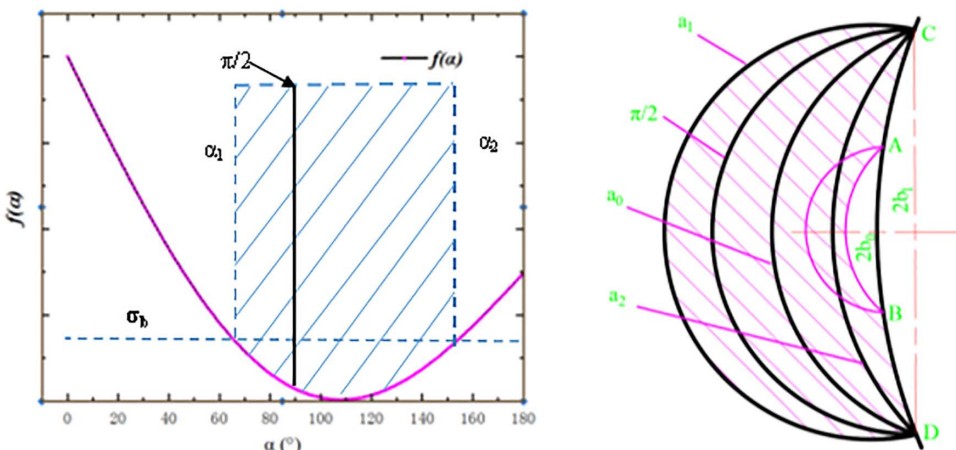

**Fig 7. Plot of function f(α) versus ultimate stress σ_b (Damage Line).**

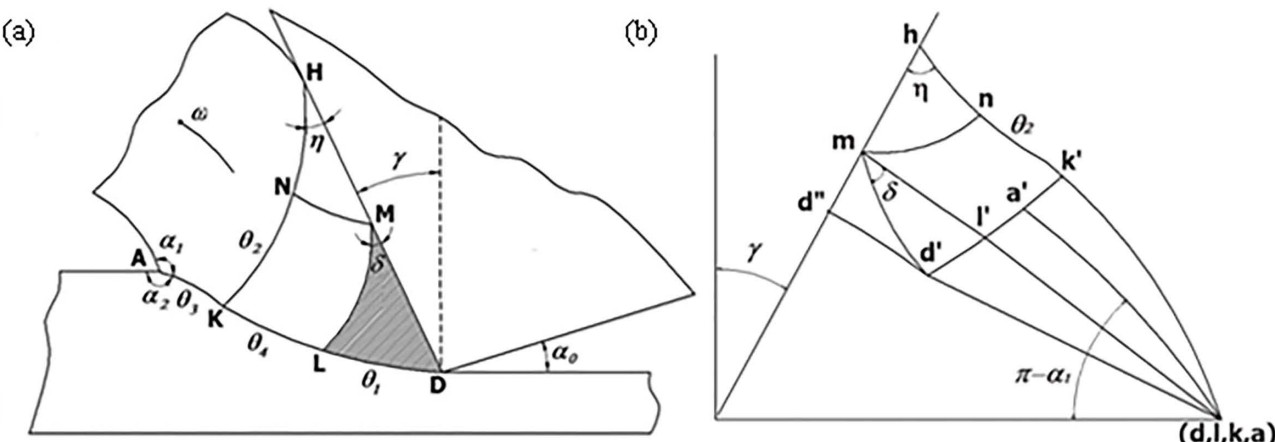

**Fig 8. Plot of function $f(\alpha)$ versus ultimate stress $\sigma_b$(Damage Band).**

Where: $a$ is the half-width of the contact between the Rounded Edge and the workpiece and $a = R\sin\gamma$

It can be seen that when a larger the arc radius of the dead metal zone is larger, the dead metal zone area is also larger; by the relationship between a and the rake angle can be seen that the larger the rake angle is the larger, and a maximum does not exceed the radius of the rounded edge. It further indicates that the size of the dead metal zone does not depend on the cutting conditions, but only depends on the rake angle, obtuse radius, and workpiece material properties.

When the mode angle is $\alpha_1$, the maximum value of the contact arc to Rounded Edges

$$d = \frac{a}{\tan\frac{\alpha_1}{2}} - R + \sqrt{R^2 - a^2}$$

(29)

The divergence point of the material is at the maximum shear stress at blunt circular contact, and according to the Mises yield criterion, at blunt circular contact

$$\tau_{xy} = \frac{[\sigma]}{\sqrt{3}}$$

(30)

At this time the shear stress reaches the maximum. According to the rounded edge-rake face, shear stress in the right-angled coordinate system can be calculated to get

$$\tau_{\max} = \frac{[\sigma]}{\sqrt{3}} = bp_0 \frac{x^2 z^2}{(x^2 + z^2)^2}$$

(31)

Where: $b = R - R\cos\gamma$, R is the radius of the rounded edge, γ is the rake angle, and z is the radius of the outer arc of the dead metal zone at the limit angle.

$$\begin{cases} p_0 = \sqrt{\frac{F}{\pi^2 R L E^*}} \\ E^* = \frac{1-\mu^2}{\pi E} \end{cases}$$

(32)

$F$ is the rounded edge contact cutting force, $\mu$ is the Poisson's ratio, $E$ is the modulus of elasticity and $L$ is the length of the workpiece.

Therefore, when the maximum cutting force during cutting, the tool radius of the rounded edge, the rake angle, and the workpiece material parameters are known to determine the stagnation point of the material.

## 3. Research methodology

### 3.1. Solidworks secondary development to solve the slip line field

The secondary development of SolidWorks is realized through its Application Programming Interface (API), which is used for some lengthy automated design work, aiming at improving productivity and optimizing the automation process. Due to the design of a large number of formulas and boundary settings in the process of solving the slip line field and drawing graphics, the use of programming software such as MATLAB and Python is required to carry out more complex arithmetic processes and can not intuitively see the process of graphical drawing, in the debugging of the dead metal zone slip line, the Rounded Edgeslip line and the Rounded Edge- the rake face of the tool slip line can not quickly find the problem, resulting in longer debugging cycle. Tool cutting process can be seen as a ramp compression plus torque process, the first elastic deformation, when the axial shear stress is greater than the maximum yield strength of the material begins to occur plastic deformation, the whole process is the first rounded edge contact, with the workpiece feed tool rake face began to contact the mechanics of the two processes and its complexity of the changes in the debugging process at this stage is most prone to errors. The use of API interface can use SolidWorks comes with the macro module to establish the basic program framework, in the debugging interface by importing the Rounded contact stress and Rounded Edge- rake face contact stress formula, the basic stress field of the component, by the formula (13) and the formula (14) in the sketch of the slip line and elasticity-plasticity demarcation line, respectively, to measure the area of the elastic zone and the plastic zone, and to calculate the proportion of the area of the plastic zone to the total area, and thus the proportion of the plastic zone to the total area, and then to use this as the basis for the commissioning process. the proportion of the plastic zone area to the total area, and then use this to reflect the magnitude of the cutting force and draw the slip line field. Therefore, this paper establishes a framework for solving the slip line field with multiple force fields and multiple boundaries on SolidWorks, and the basic flow is shown in Fig 9.

### 3.2. Simulation

Through the introduction we know that a large number of scholars have observed the stagnation point and dead metal zone through simulation, this paper adopts three-factor four-level orthogonal cutting to simulate the cutting processing, the purpose is to observe the stagnation point and dead metal zone through simulation and verify the theoretical model established in the previous period, the specific simulation parameters are shown in Table 1.

Modeling is done by orthogonal cutting, the tool is selected from the library with its carbide tool, the size is set according to Table 1, the workpiece is selected from the library with its own TC4, the size is set to width (5 mm) * height (10 mm), the tool mesh is divided into 700 cells, the workpiece mesh is divided into 25 cells, the minimum number of steps is set to 10, the friction coefficient is set to 0.2, and the damage model used is the J-C model.

$$\sigma = (A + B\varepsilon^n)\left(1 + C\ln\frac{\dot{\varepsilon}}{\dot{\varepsilon}_0}\right)\left(1 - \left(\frac{T - T_{room}}{T_{melt} - T_{room}}\right)^m\right)s^{-1}$$

(33)

Where: A is the initial yield stress in MPa; B is the strain hardening constant in MPa; C is the coefficient of identity; m is the thermal softening coefficient; and n is the cutting hardening index.

Table 2 shows the JC parameters of TC4 titanium alloy.

Fig 10 shows the simulation post-processing 2D cutting diagram (maximum stress state).

Cutting simulation is carried out through modeling and setting of relevant parameters, and the corresponding simulation results are extracted in the corresponding interface after the simulation is completed, and the cutting schematic diagrams

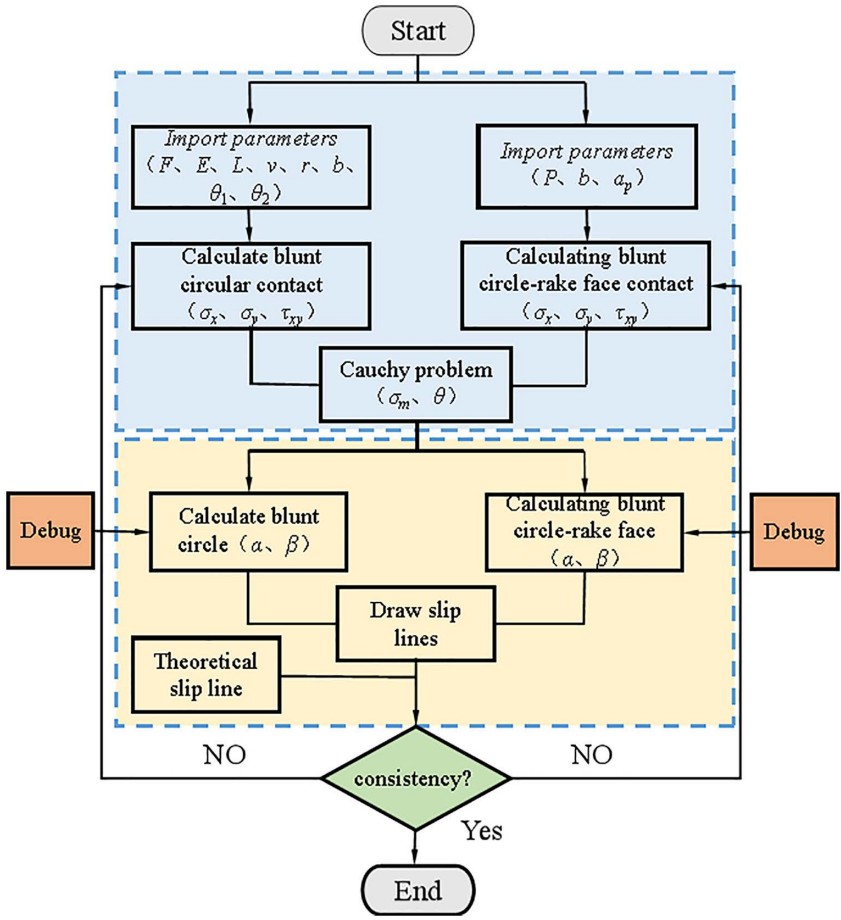

**Fig 9. Calculation process for secondary development of the slip line field.**

in the speed condition and equivalent effect variation are extracted respectively so that the dead-zone stagnation points in the cutting process can be directly observed.

### 3.3. Orthogonal cutting experiment

This experiment uses orthogonal cutting to verify the slip-line field model. Firstly, the tool fixture and workpiece fixture are installed on the AGS-X electronic universal testing machine, and the installation is shown in Fig 11; secondly, the high-speed camera and searchlight are installed to complete the focusing of the light and the high-speed camera; finally, the relevant data of the cutting process are uploaded to the computer through the sensor connector of the AGS-X electronic universal testing machine.

The equipment used in this cutting experiment is shown in Table 3:

## 4. Results and analysis

### 4.1. Stagnation point

The material in the cutting process, with the continuous feed, will be cut into two parts that are, chips and residual material, this is due to the cutting material being torn by the cutting tool under the action of the formation of slip

**Table 1. Three factors and four levels of orthogonal cutting simulation.**

| Simulation Number | Rake angle (°) | Rounded edge (mm) | Depth (mm) | Speed (m/min) | Feed rate (mm/r) | Relief angle (°) |
|---|---|---|---|---|---|---|
| 1 | 5 | 0.05 | 0.1 | 15 | 0.1 | 5 |
| 2 | 10 | 0.1 | 0.1 | 15 | 0.1 | 5 |
| 3 | 15 | 0.15 | 0.1 | 15 | 0.1 | 5 |
| 4 | 20 | 0.2 | 0.1 | 15 | 0.1 | 5 |
| 5 | 10 | 0.05 | 0.1 | 15 | 0.2 | 5 |
| 6 | 5 | 0.1 | 0.1 | 15 | 0.2 | 5 |
| 7 | 20 | 0.15 | 0.1 | 15 | 0.2 | 5 |
| 8 | 15 | 0.2 | 0.1 | 15 | 0.2 | 5 |
| 9 | 15 | 0.05 | 0.1 | 15 | 0.3 | 5 |
| 10 | 20 | 0.1 | 0.1 | 15 | 0.3 | 5 |
| 11 | 5 | 0.15 | 0.1 | 15 | 0.3 | 5 |
| 12 | 10 | 0.2 | 0.1 | 15 | 0.3 | 5 |
| 13 | 20 | 0.05 | 0.1 | 15 | 0.4 | 5 |
| 14 | 15 | 0.1 | 0.1 | 15 | 0.4 | 5 |
| 15 | 10 | 0.15 | 0.1 | 15 | 0.4 | 5 |
| 16 | 5 | 0.2 | 0.1 | 15 | 0.4 | 5 |
| 17 | 5 | 0.05 | 0.2 | 15 | 0.1 | 5 |
| 18 | 5 | 0.05 | 0.3 | 15 | 0.1 | 5 |
| 19 | 5 | 0.05 | 0.4 | 15 | 0.1 | 5 |
| 20 | 5 | 0.05 | 0.5 | 15 | 0.1 | 5 |
| 21 | 5 | 0.05 | 0.6 | 15 | 0.1 | 5 |
| 22 | 10 | 0.05 | 0.1 | 15 | 0.1 | 5 |
| 23 | 15 | 0.05 | 0.1 | 15 | 0.1 | 5 |
| 24 | 20 | 0.05 | 0.1 | 15 | 0.1 | 5 |
| 25 | 25 | 0.05 | 0.1 | 15 | 0.1 | 5 |

**Table 2. Initial parameters of the J-C model for TC4Alloy [29].**

| A | B | n | C | m |
|---|---|---|---|---|
| 800 | 832 | 0.42 | 0.046 | 1.3 |

on the formation of chips, the speed direction for the direction of the rake face, and with the horizontal axis of the material will be symmetrical direction of the cutting tool "ironing" to ensure that the surface quality of the material. The material will be "ironed" by the tool in a direction symmetrical to the horizontal axis, thus ensuring the surface quality of the material. Cutting shunt occurs in the critical damage value of the material, according to Miss yield quasi can be calculated to get the critical value of the destruction of the material that is the formula (31), according to the different rake angle, different feed, different obtuse radius and different depth of cut will be the theoretical value of the cutting x and z were listed in the table, you can get the stagnation point coordinates (Note: x-axis over the center of the circle and the obtuse arc of a parallel division, z-axis and obtuse tangent to the circle and perpendicular to the z-axis).

In this paper, we analyze the stagnation points during the cutting process for tool rake angle, Rounded Edgeradius, and cutting feed. Firstly, according to the simulation post-processing to extract the stagnation points at different rake angles are shown in Fig 12:

 

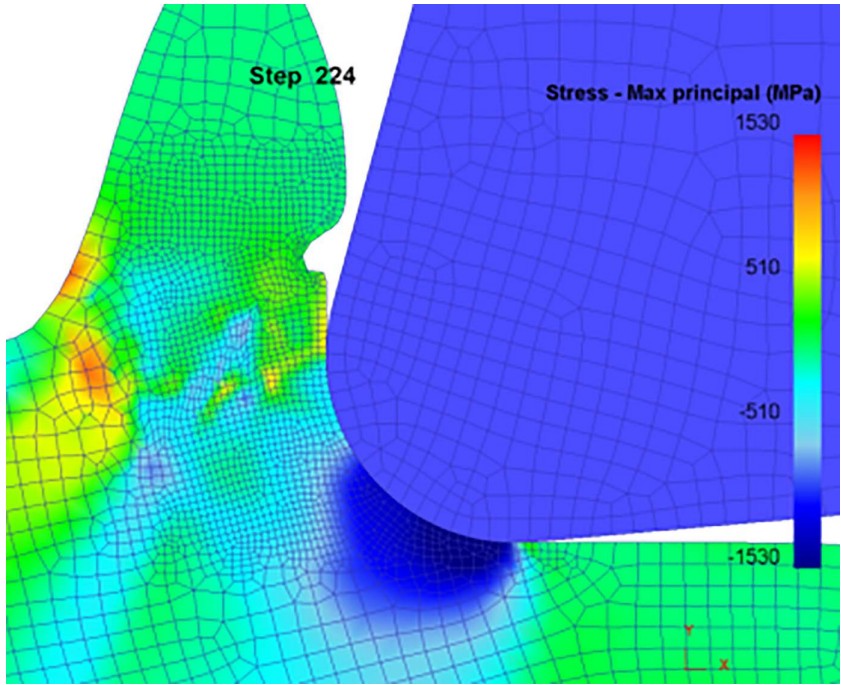

**Fig 10. 2D simulation post-processing (maximum stress state).**

From Fig 12, it can be seen that there are slight differences in the stagnation points of the machined workpieces when machining with different rake angles, mainly in the X and Z directions centered on the rounded edge. Meanwhile, we list the theoretical and simulated values of different rake angles in Table 4 according to the simulation results.

According to Table 4 can be analyzed that the tool rake angle for the cutting process of the material shunt has a great impact on the theory by the formula 31 to see the stagnation point in the X and Z direction with the increase in the rake angle increases, from the simulation results to see the verification of the correctness of the theory.

According to Eq. 28, it can be seen that the rounded edge of the tool also has a certain effect on the stagnation point, so the stagnation point is extracted in the post-processing of the simulation for different blunt radii as shown in Fig 13:

From Fig 13, it can be seen that there is a slight difference in the stagnation point of the machined workpiece in the rounded edge machining of different tools, which is mainly manifested in the X and Z directions with the rounded edge as the center. Meanwhile, we list the theoretical and simulated values of different rake angles in Table 5 according to the simulation results.

According to Table 5 the tool Rounded Edge for the cutting process material shunt for a greater impact, can be understood as with the tool Rounded increases, the tool and the workpiece contact area increases, which will make the workpiece tear less obvious, but will make the processing of the surface quality to get some improvement. According to equation 28, the larger the radius of the rounded edge, the larger the formation of the dead metal zone, which will make the stagnation point closer to the inside of the workpiece, which in turn affects the distribution of the stagnation point, the overall presentation of the distribution of 45 ° upward, which has been verified in the simulation and analysis process.

To verify that the effect of other cutting parameters on the stagnation points is not regular, the stagnation points at different feeds are extracted in the simulation post-processing as shown in Fig 14.

From Fig 14, it can be seen that the distribution of stagnation points is not uniform when machining with different feeds, so the simulation results of the theoretical and simulated values under different feed conditions are listed in Table 6.

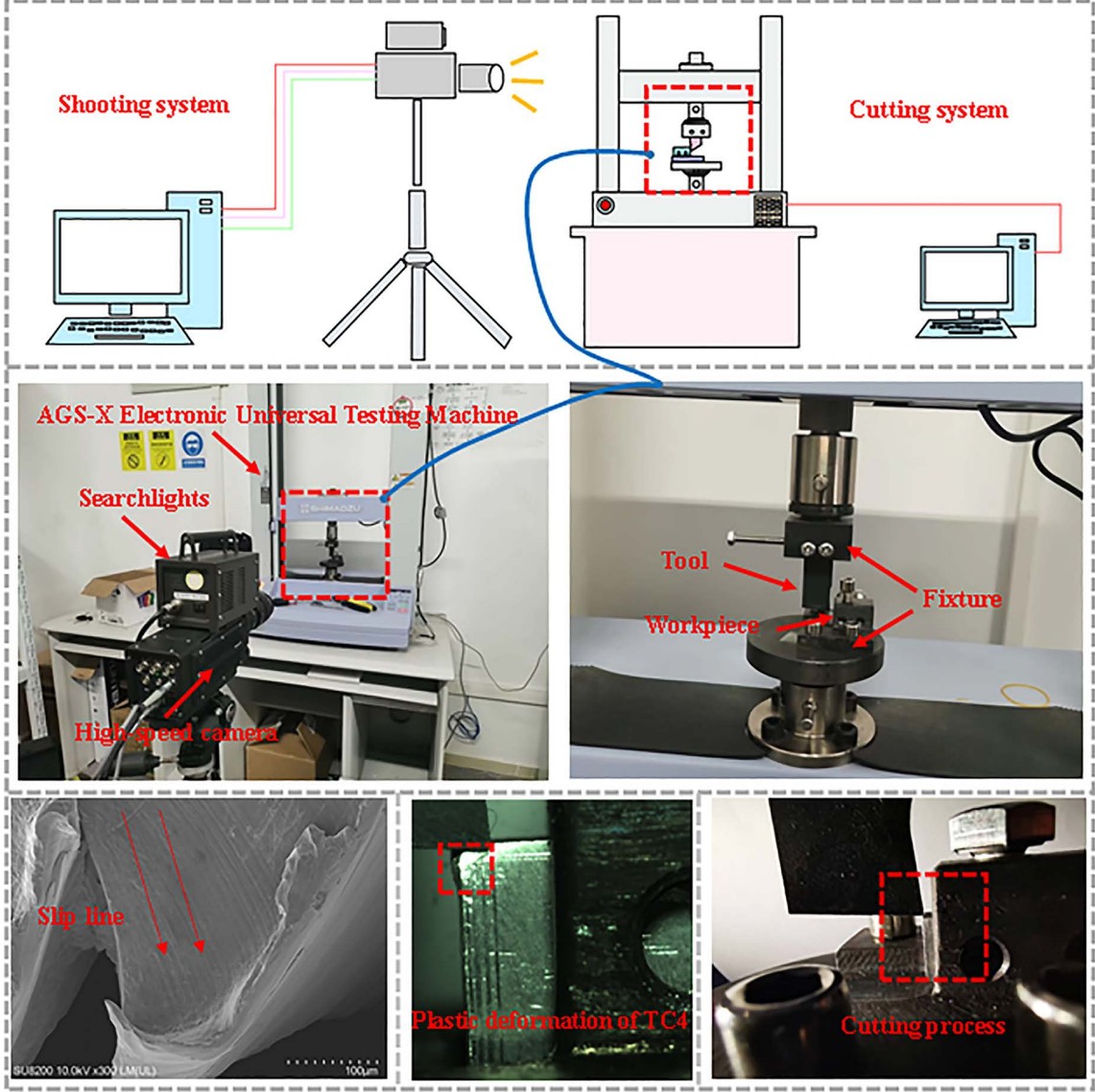

**Fig 11. Cutting experiment.**

**Table 3. Experimental devices and models.**

| Title | Model Number |
| --- | --- |
| Compactors | AGS-X |
| Tool | M2 co5 66–69 |
| High-speed camera | MotionXtraO9-S1 |

According to Table 6, it can be obtained that the influence of tool feed on the stagnation point is not regular, which is further confirmed in the simulation. It can be shown that the main factors affecting the stagnation point are still the tool rake angle and the tool radius of the rounded edge, and the influence of other cutting parameters on the stagnation point

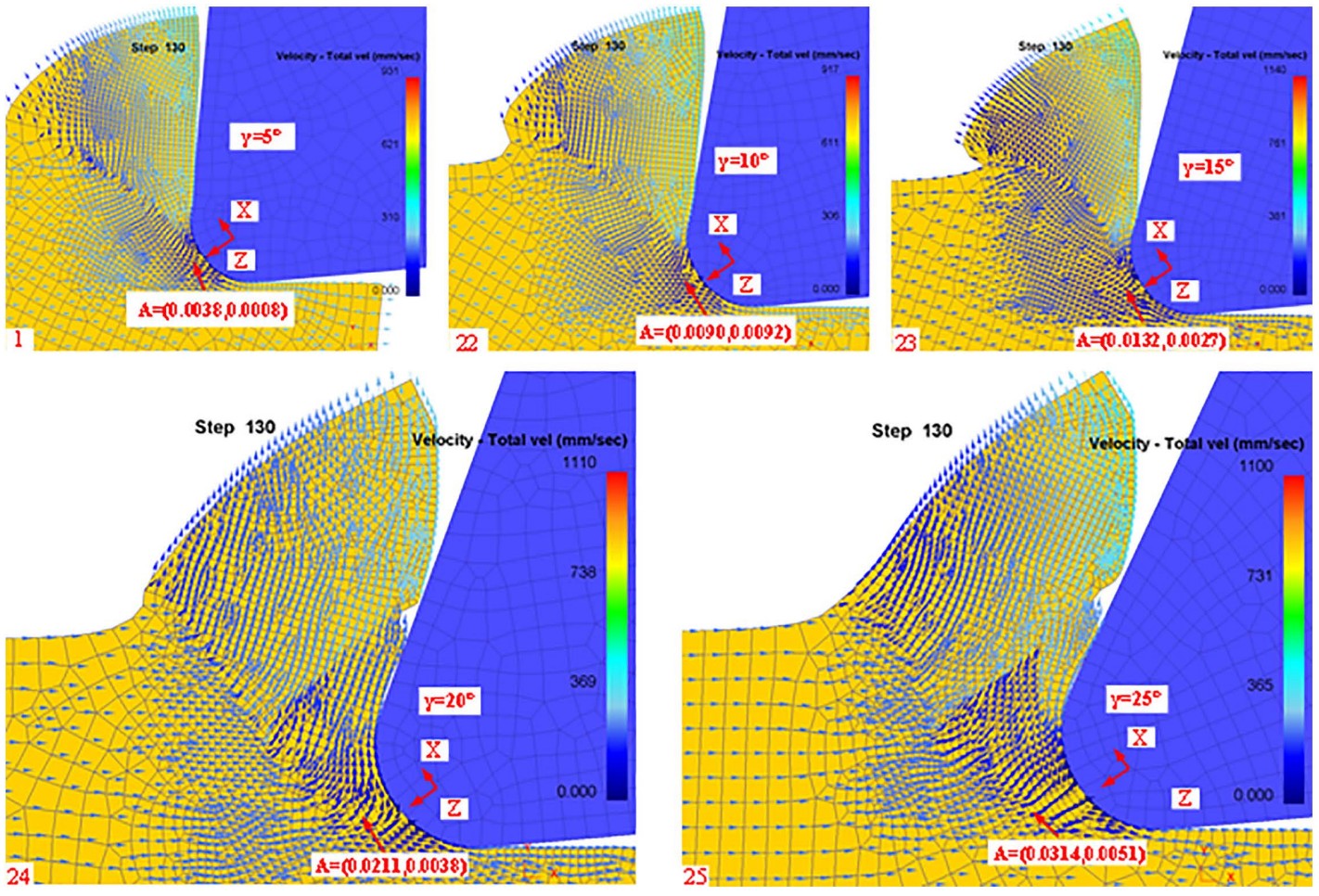

**Fig 12. Post-simulation processing of stagnation points (different rake angles).**

**Table 4. Theoretical coordinates of the diversion point (effect of Rake angle).**

| Simulation Number | Rake angle (°) | Theoretical z-value (mm) | Theoretical x-value (mm) | Simulation z-value (mm) | Simulation x-value (mm) |
|---|---|---|---|---|---|
| 1 | 5 | 0.0046 | 0.0009 | 0.0038 | 0.0008 |
| 22 | 10 | 0.0091 | 0.0018 | 0.0090 | 0.0092 |
| 23 | 15 | 0.0136 | 0.0028 | 0.0132 | 0.0027 |
| 24 | 20 | 0.0180 | 0.0036 | 0.0211 | 0.0038 |
| 25 | 25 | 0.0222 | 0.0044 | 0.0314 | 0.0051 |

can be ignored. To more intuitively describe the coordinate distribution of different rake angles, obtuse radii, and feeds in the machining process, the corresponding stagnation points are plotted in Figs 15–17.

Figs 15 and 16 show that: the theoretical derivation of the rake angle and the blunt radius of the tooltip results in the distribution of the stagnation point and the actual simulation results in the same trend and distribution, while Fig 17 shows that: the effect of cutting feed on the stagnation point does not show a regular change, and there is a large error between the simulation value and the theoretical value.

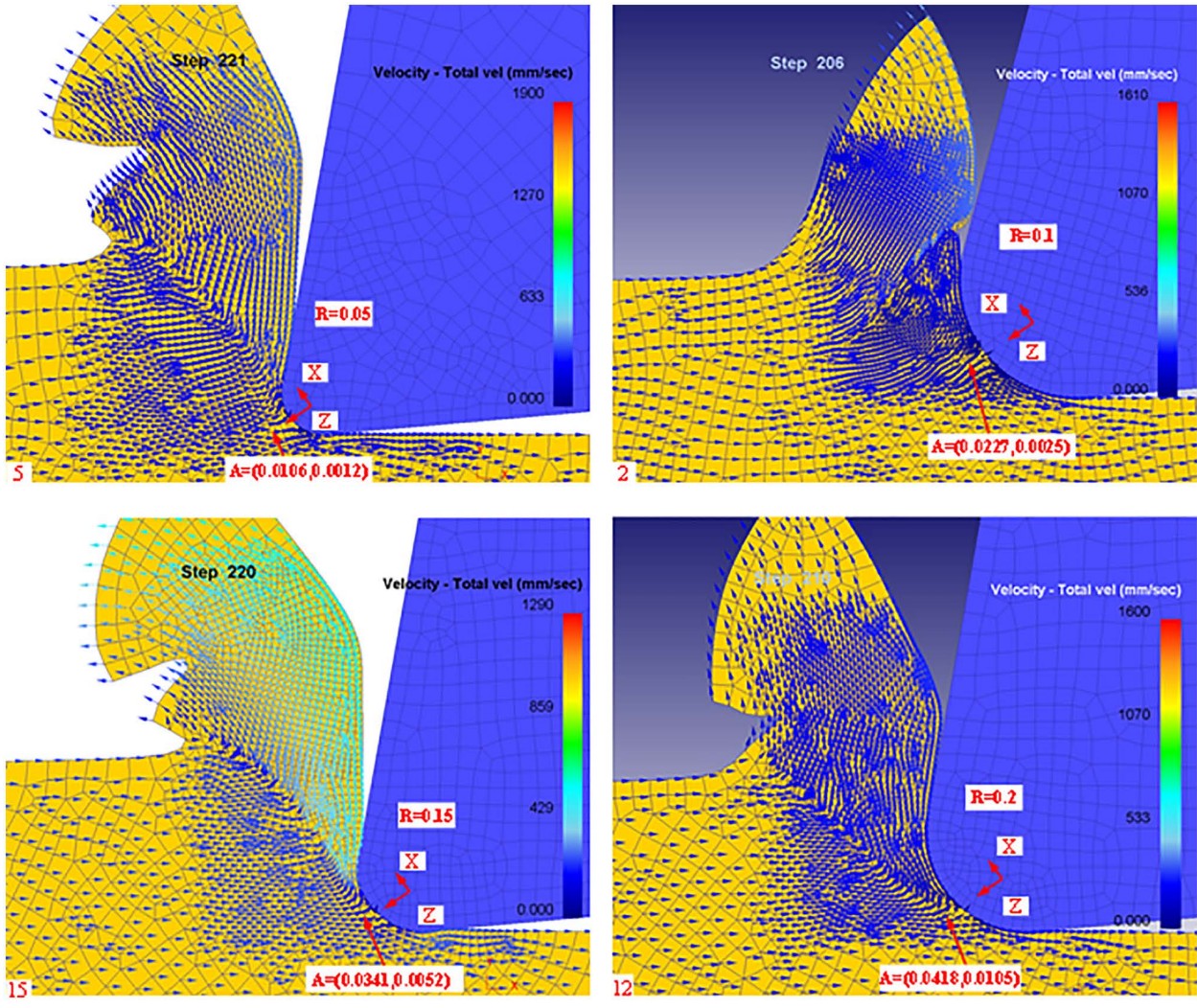

**Fig 13. Diversion points (different Blunt radius of the tip).**

**Table 5. Theoretical coordinates of the stagnation point (effect of the Rounded edge).**

| Simulation Number | Rounded edge (mm) | Theoretical z-value (mm) | Theoretical x-value (mm) | Simulation z-value (mm) | Simulation x-value (mm) |
|---|---|---|---|---|---|
| 5 | 0.05 | 0.0091 | 0.0018 | 0.0106 | 0.0012 |
| 2 | 0.10 | 0.0183 | 0.0028 | 0.0227 | 0.0025 |
| 15 | 0.15 | 0.0272 | 0.0048 | 0.0341 | 0.0052 |
| 12 | 0.20 | 0.0365 | 0.0063 | 0.0418 | 0.0105 |

## 4.2. Dead metal zone

In the cutting process there will be a material accumulation phenomenon under certain conditions to form a dead metal zone, in many papers in the simulation process observed the existence of the dead metal zone, based on the simulation in the velocity interface to observe the dead metal zone morphology, while the observation of elastic-plastic deformation

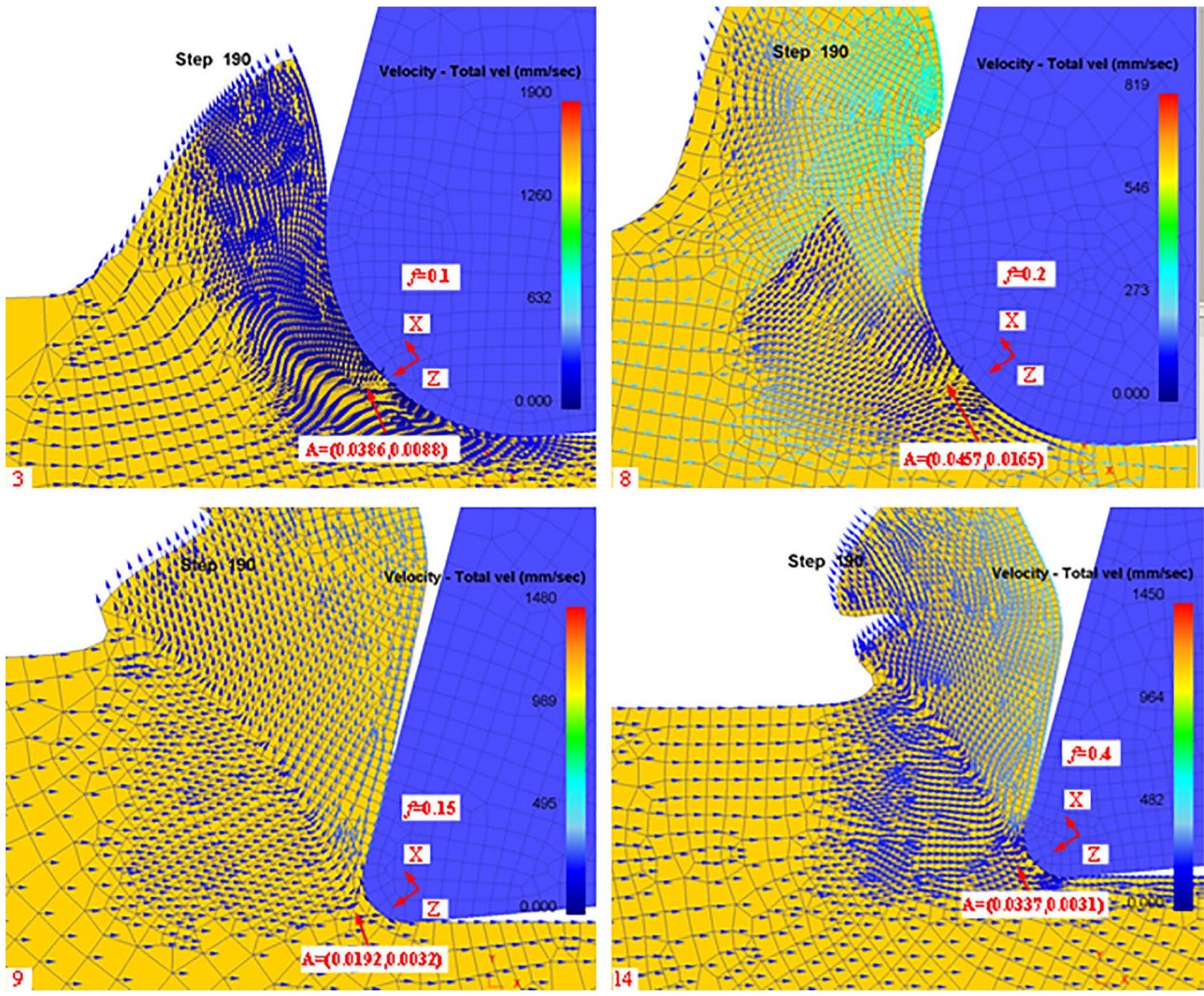

**Fig 14. Stagnation point (different feeds).**

**Table 6. Theoretical coordinates of the stagnation point (effect of different feeds).**

| Simulation Number | Feed (mm/r) | Theoretical z-value (mm) | Theoretical x-value (mm) | Simulation z-value (mm) | Simulation x-value (mm) |
|---|---|---|---|---|---|
| 3 | 0.1 | 0.0408 | 0.0076 | 0.0386 | 0.0088 |
| 8 | 0.2 | 0.0544 | 0.0103 | 0.0457 | 0.0165 |
| 9 | 0.3 | 0.0136 | 0.0028 | 0.0192 | 0.0032 |
| 14 | 0.4 | 0.0272 | 0.0048 | 0.0337 | 0.0031 |

zone demarcation line. The parameters selected from Table 1 are simulated, and the dead metal zone morphology under different parameters is intercepted as shown in Fig 18:

Such as Fig 18 can be observed under different cutting conditions of the dead metal zone morphology, as the cutting speed increases the cutting force gradually decreases, and the blue part of the Fig 18 represents a relatively slow cutting

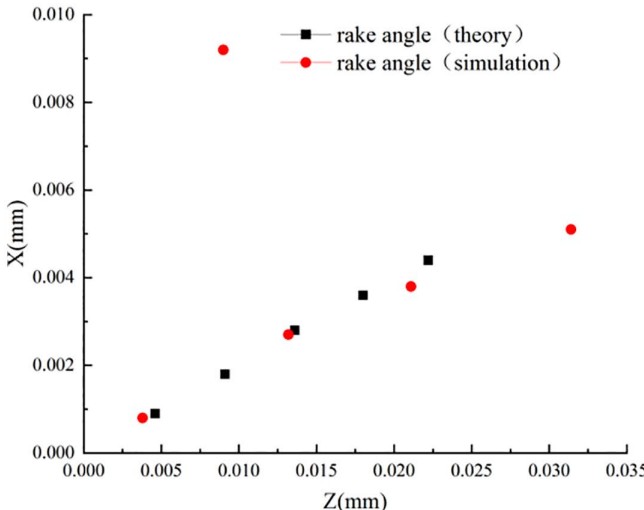

**Fig 15. Distribution of stagnation points at different rake angles.**

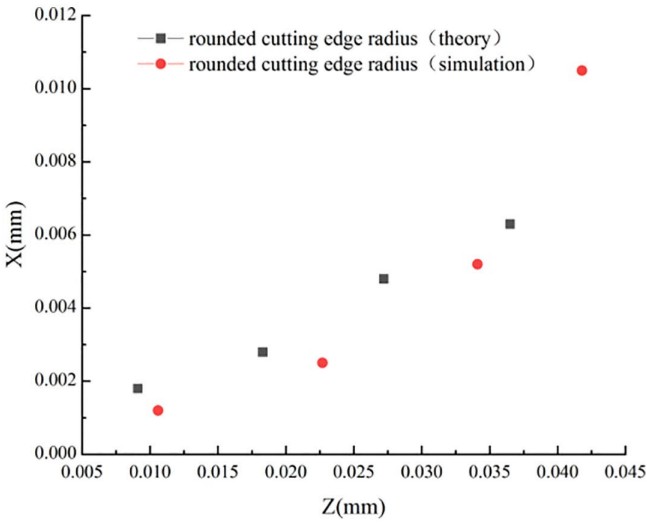

**Fig 16. Distribution of stagnation points for different rounded edge.**

speed generated by the cutting force and relatively large, so it can be seen that the part of the material accumulation that is the dead metal zone. However, with continuous feeding, the dead metal zone also changes, this is due to the material at the stagnation point respectively upward and downward flow with the formation of chips and causing certain changes, the size of the dead metal zone with the shunt material to take away the amount of material and change. Velocity changes in the region for the relatively small force, and the region is the most obvious plastic deformation occurs in the region, indicating that the region for the main deformation zone, and the cause of this phenomenon is the tool on the material will produce cracks, and then based on the theory of the slip line field on the formation of plastic deformation. The slip line field is based on contact stress derivation, so different velocity curves can be regarded as different iso-stress curves, which can be seen from the elastic deformation to plastic deformation process during cutting, which explains the stress derivation of blunt circular contact very well.

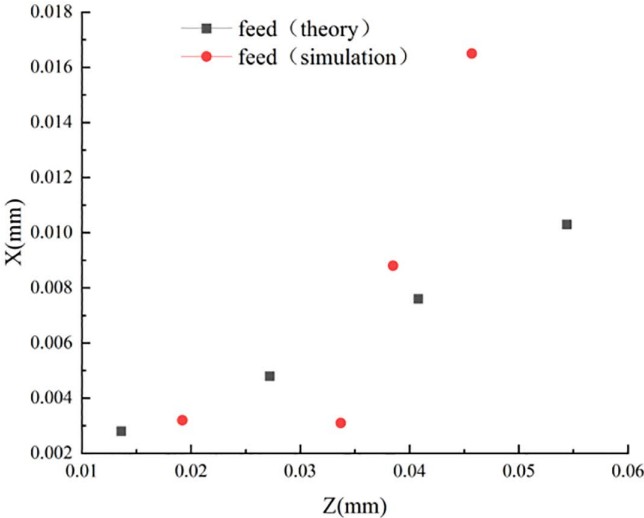

**Fig 17. Distribution of stagnation points at Different Feeds.**

The dead metal zone outside the boundary of the arc radius formula can be known as the arc radius of the dead metal zone and the tool rake angle, Rounded Edgeradius and the occurrence of the angle of the damage zone, and the same material damage zone is only related to the intrinsic properties of the workpiece material, this part can not be derived through the relevant theoretical modeling for the time being, but can be assumed that the maximum damage zone of the angular value (defined as the modal angle) to analyze the rule of change of the dead metal zone. In this paper, we analyze the dead metal zone in the cutting process for the tool rake angle, Rounded Edgeradius, and cutting feed. First of all, according to the simulation post-processing to extract the dead metal zone at different rake angles is shown in Fig 19:

The simulated dead metal zones with rake angles of 5°, 10°, 15°, and 20° are shown schematically in Fig 16, respectively, and the dead metal zone region is outlined according to the maximum stress region, and the simulated limit angle and mode angle are measured, and the values between the theoretical maximum limit angle obtained by Eq. 29 to the Rounded Edge and the simulated measured values are listed in Table 7.

From the theoretical and simulation results, it can be seen that as the lead angle increases the outer arc radius of the dead metal zone also increases, indicating that the dead metal zone also increases and the error between the simulation value and the theoretical value is small, and at the same time can be analyzed and concluded that the simulation of the average value of the mode angle is about 78 °. Therefore, in the actual machining process, to avoid the generation of the dead metal zone or reduce the accumulation of the dead metal zone, you can use a small rake angle tool for cutting. According to Eq. 29, it can be seen that the rounded edge of the tool also has a certain effect on the maximum diameter of the arc outside the dead metal zone, so the dead metal zone is extracted in the simulation post-processing with different rounded edge radii as shown in Fig 20:

The simulated dead metal zones with Rounded Edgeradii of 0.05 mm, 0.1 mm, 0.15 mm, and 0.20 mm are shown in Fig 20, respectively, and the dead metal zone region is outlined according to the maximum stress region, and the simulated limit angle and mode angle are measured to obtain the simulated limit angle and the mode angle and the distance of the dead metal zone arcs to the Rounded Edgeare listed in Table 8 at the same time.

From the theoretical and simulation values, it can be seen that with the increase of the radius of the rounded edge, the outer arc radius of the dead metal zone also increases, the theoretical and actual value of the error is small, and the die angle with the different rake angle as mentioned above also tends to be about 78 °. Therefore, in the actual machining

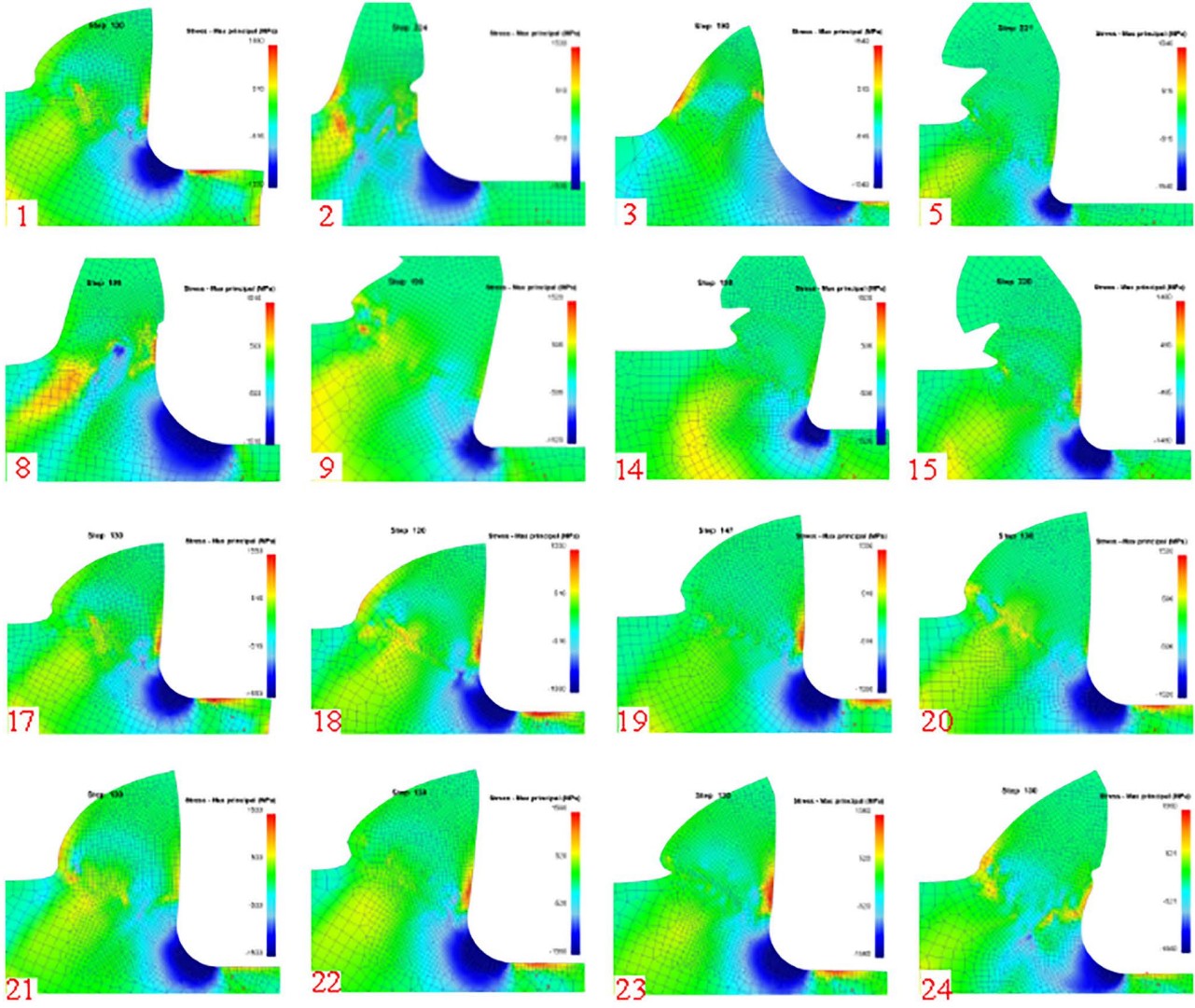

**Fig 18. Dead metal zone morphology for different cutting parameters.**

process, to avoid the generation of dead metal zones or reduce the accumulation of dead metal zones, can be used in a small rake angle small rounded edge tool for cutting.

To verify that the effect of other cutting parameters on the dead metal zone is not regular, the stagnation points at different feeds are extracted in the simulation post-processing as shown in Fig 21.

The simulated dead metal zones with feeds of 0.1 mm, 0.2 mm, 0.3 mm, and 0.4 mm are shown in Fig 21, respectively, and the dead metal zone area is outlined according to the maximum stress area, and the simulated limit angle and die angle are measured, and the die angle and the distance from the dead metal zone arc to the rounded edge are listed in Table 9.

Comparing the theoretical and simulation values, it can be seen that the outer arc radius of the dead metal zone does not show a regular change with the increase of feed, and there is a large error between the simulation and the theoretical values, but the mode angle is also close to about 78° as the different rake angles mentioned above.

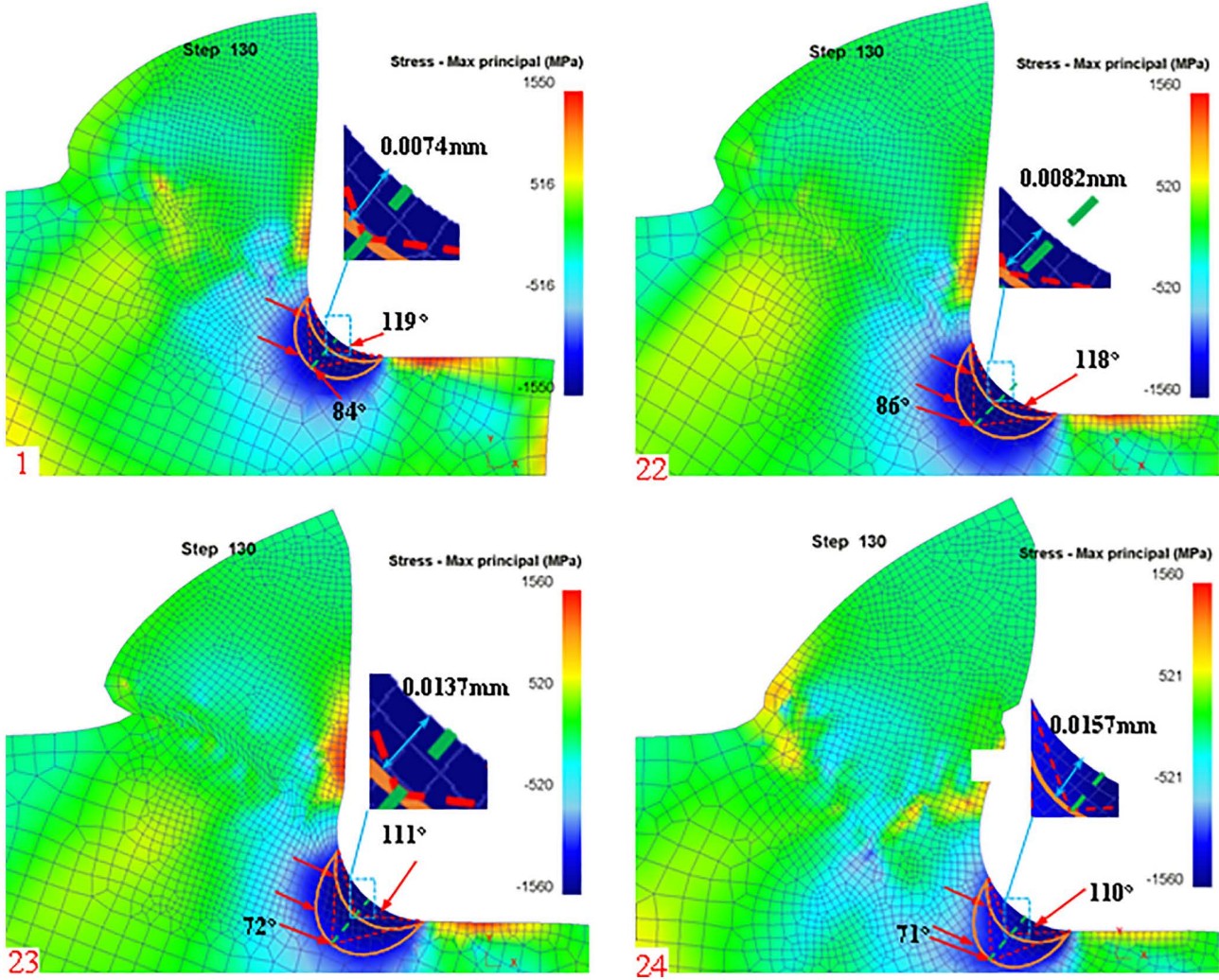

**Fig 19. Dead metal zones for different rake angles.**

**Table 7. Dead metal zone parameters at different rake angles.**

| Simulation Number | Rake angle (°) | Simulated mode angle(°) | Theoretical dead metal zone – Rounded Edgelength(mm) | Simulation Dead metal zone – Rounded EdgeLength(mm) |
|---|---|---|---|---|
| 1 | 5 | 84 | 0.0052 | 0.0074 |
| 22 | 10 | 86 | 0.0100 | 0.0082 |
| 23 | 15 | 72 | 0.0143 | 0.0137 |
| 24 | 20 | 71 | 0.0181 | 0.0157 |

This section through the simulation of the dead metal zone outside the arc mode angle tool rake angle and rounded edge radius shows a positive correlation, but there are shortcomings through a series of derivations still can not get the arc mode angle formula, but through the simulation of the angle can be seen in the simulation is about 78 ° or so, which will be the further deduction of the validation of the work we will do later.

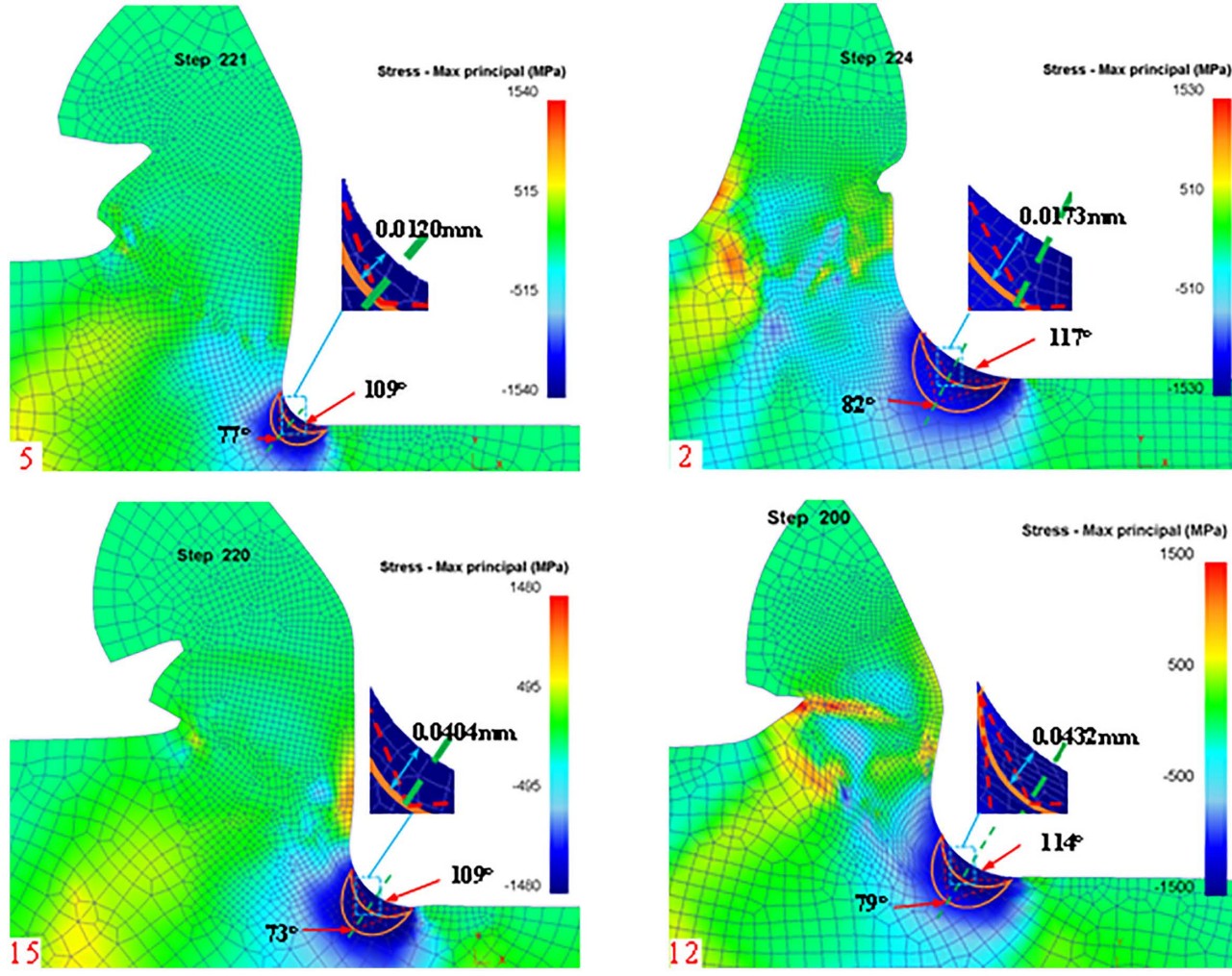

**Fig 20. Dead metal zone for different blunt radii of the tooltip.**

**Table 8. Dead metal zone parameters for a different rounded edge.**

| Simulation Number | Rake angle (°) | Simulated mode angle(°) | Theoretical dead metal zone – Rounded Edgelength(mm) | Simulation Dead metal zone – Rounded EdgeLength(mm) |
|---|---|---|---|---|
| 5 | 0.05 | 77 | 0.0100 | 0.0120 |
| 2 | 0.10 | 82 | 0.0199 | 0.0173 |
| 15 | 0.15 | 73 | 0.0298 | 0.0404 |
| 12 | 0.20 | 79 | 0.0398 | 0.0432 |

## 4.3. Slip line modeling analysis

Through the secondary development in Solidworks, calculate and draw the slip line field of different rake angles: based on the slip line field formed by different cutting rake angles, the change of slip line field will lead to the change of contact length with the change of the corresponding rake angle, which will lead to the different slip field. Now through the SolidWorks

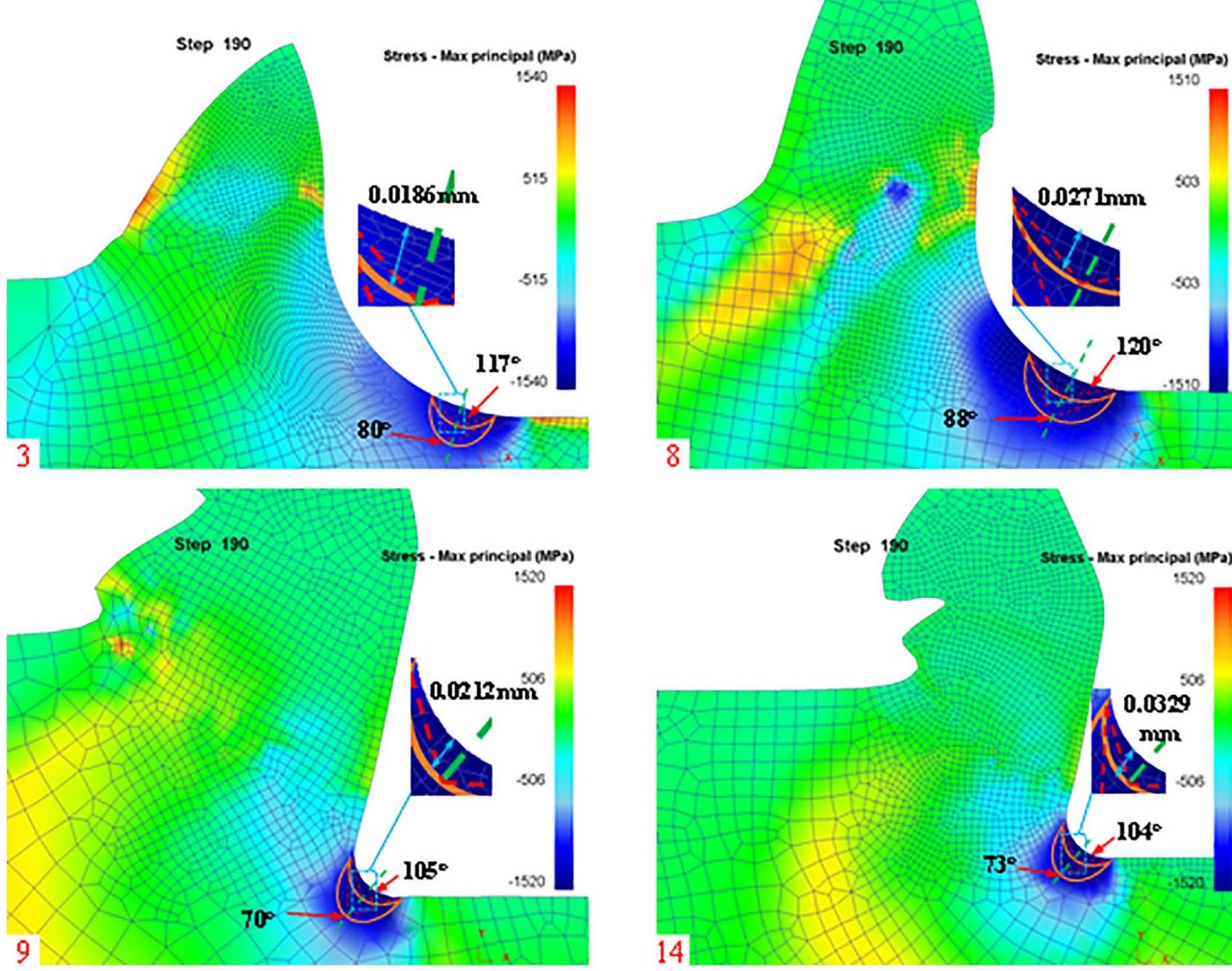

**Fig 21. Dead metal zone for different feeds.**

**Table 9. Dead metal zone parameters at different feeds.**

| Simulation Number | Feed (mm/r) | Simulated mode angle (°) | Theoretical dead metal zone – Rounded Edgelength(mm) | Simulation Dead metal zone – Rounded EdgeLength(mm) |
|---|---|---|---|---|
| 3 | 0.1 | 80 | 0.0428 | 0.0186 |
| 8 | 0.2 | 88 | 0.0517 | 0.0271 |
| 9 | 0.3 | 70 | 0.0142 | 0.0212 |
| 14 | 0.4 | 73 | 0.0286 | 0.0329 |

secondary development for different rake angle slip line field change rule research, derive the back angle of 5 °, r = 0.1 mm, depth of cut = 0.1 mm, rake angle of 5 °, 10 °, 15 °, 20 ° and 25 ° of the slip line field as shown in the Fig: through the second development in SolidWorks, calculation and drawing of different rake angle of the slip Line field: Based on the slip line field formed by different cutting angles, the change of slip line field will lead to the change of contact length with the change

of the corresponding rake angle, which will lead to the difference of the slip field. Now through the SolidWorks secondary development for different rake angle slip line field change rule research, derive the back angle of 5 °, r=0.1 mm, depth of cut=0.1 mm, the rake angle of 5 °, 10 °, 15 °, 20 ° and 25 ° of the slip line field as shown in Fig 22:

As shown in Fig 22 ABD is the contact surface of the rake face of the tool, by the theoretical derivation of the second chapter of the A point stress is the largest, D point stress minimum, and wireless tends to zero; due to the increase in the rake angle, the tool rounded edge and the workpiece contact length increases, which makes the first deformation of the area of the region of the percentage increase in the area of the third deformation of the region of the area of the proportion of the workpiece and the rake face of the contact length decreases and reduces the cutting force in the rake angle of the increase will make the cutting force decreases. The increase of cutting force in the rake angle will make the cutting force decrease, this is due to the first deformation zone affecting the size of the cutting force of the main slip line area, as the rake angle becomes larger, the rounded edge area contact length becomes longer, it will lead to the contact area of the rake face to reduce, but the rounded edge area to increase the length of the contact is less than the length of the rake face to reduce the length of the rake face, so the cutting force is with the rake angle increases and decreases.

Slip line field at different depths of cut: Based on the slip line field formed at different depths of cut, the change in the slip line field will result in a change in the direction of the chip formation during cutting (the angle is getting bigger and bigger) along with the corresponding change in depth of cut, which in turn leads to the difference in the slip field. The derived back angle of 5 °, r=0.1 mm, rake angle=5 °, depth of cut of 0.1 mm, 0.2 mm, 0.3 mm, 0.4 mm, 0.5 mm, and 0.6 mm slip line field as shown in Fig 23:

As shown in Fig 23, the highest point of the tool is the center, the formation of two sides of the symmetry of the slip zone, at this time only the rounded edge formed by the slip line field, with the increase in the depth of cut of the contact length of its rake face is also increasing, while the plastic deformation zone is also becoming larger.

Slip line field of different tools blunt radius: blunt radius plays a crucial role in the cutting process, with the increase of blunt radius cutting force will increase, while the blunt contact length will also increase, which makes the elastic deformation during cutting more intense, which leads to the deformation of the range increases, by importing the relevant parameters to get the blunt radius of 0.05 mm and 0.1 mm when the slip line field, the blunt radius of 0.1 mm and 0.1 mm. As shown in Fig 24:

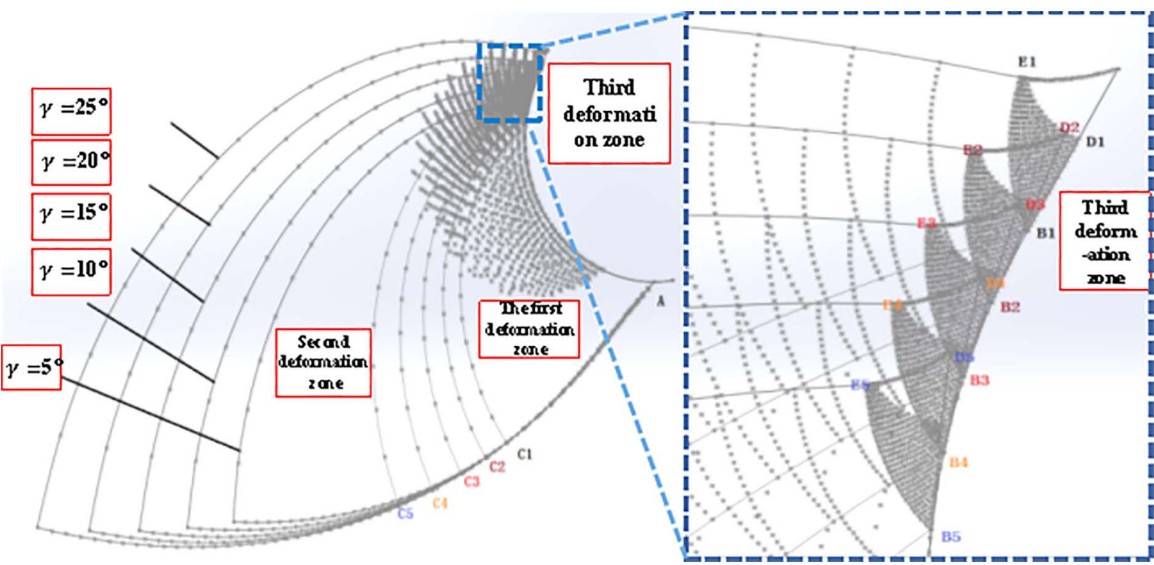

**Fig 22. Slip line field for different rake angles.**

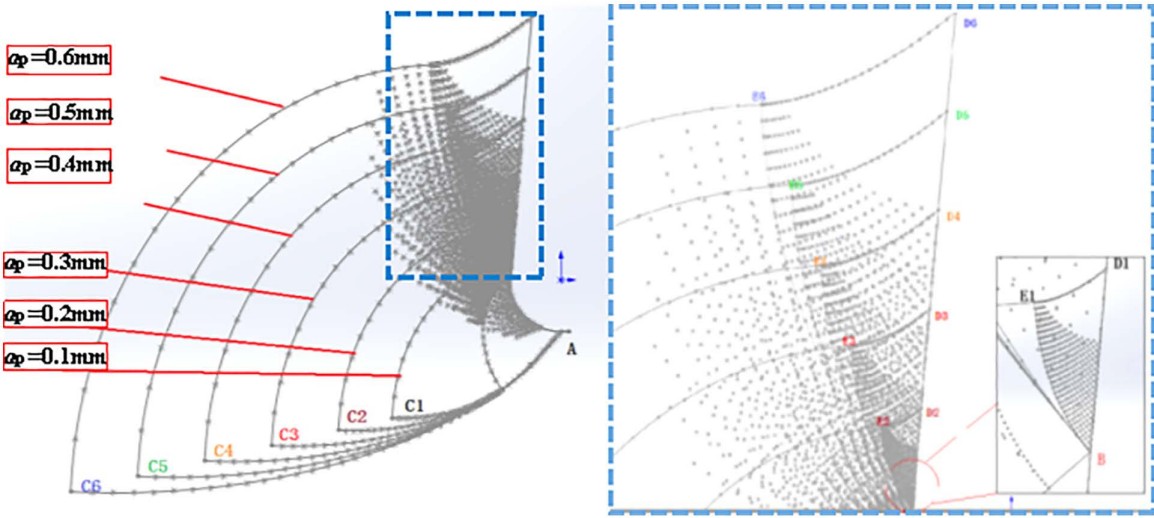

**Fig 23. Slipline field with different depths of cut (a) Slipline field; (b) Slipline in the third deformation zone.**

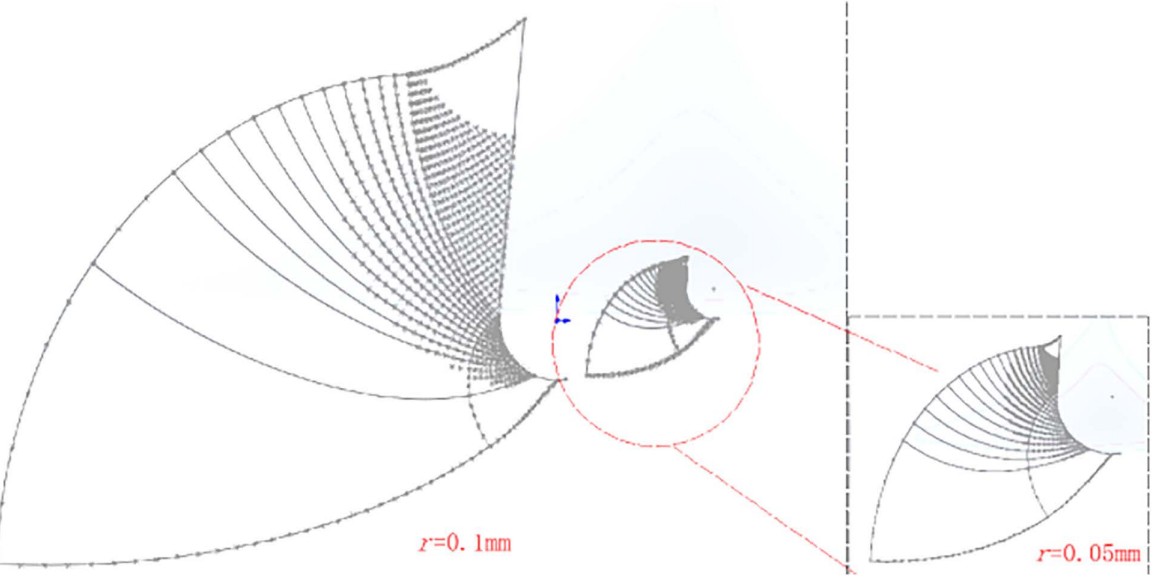

**Fig 24. Slip line field for different rounded edge radius.**

As shown in Fig 24 the contact length of the front tool face is denoted as $L_a$, and the contact length of the arc segment for the arc is denoted as $L_b$, which can be derived as the contact length:

$$L_a = \frac{4P}{\pi E \sin \gamma} - r \tag{34}$$

$$L_b = r(\varphi + \gamma) \tag{35}$$

$$\varphi = \arcsin\frac{b}{r} \tag{36}$$

The total contact length is:

$$L = L_a + L_b = r(\varphi + \gamma - 1) + \frac{4P}{\pi E \sin \gamma} \tag{37}$$

From Eq. 37, it can be seen that the contact length and the radius of the rounded edge are positively proportional to each other.

The dead metal zone of the slip line field: based on the parameters depth of cut $a_p = 0.4$ mm, rake angle 25°, rounded edge radius 0.05 mm, can be obtained by the program as shown in Fig 25:

As shown in Fig 25 CDE is the contact surface of the rake face of the tool, the maximum stress at point C, the minimum stress at point D, the CEFGC area for the plastic zone, and ACGA for the elastic zone. For the BH region, that is, the "dead metal zone", this part of the material is always in front of the tool and does not change, if the cutting conditions are appropriate, will form a "the built-up edge".

Orthogonal cutting experiments were carried out on the AGS-X electronic universal testing machine by using a high-speed camera to capture the cutting process based on different cutting parameters. As shown in Fig 10 is the cutting schematic with the parameters: 15° rake angle, 5° back angle, 0.15 mm rounded edge radius of the tooltip, 0.1 mm depth of cut, 15 m/min cutting speed, and 0.1 mm/ feed. The orthogonal cutting process can be observed under the high-speed camera with rounded edge contact, rounded edge-rake face contact, and steady-state cutting, to carry out orthogonal cutting more accurately, the experiment is marked in advance on the surface of the workpiece TC4 scales, which is convenient for the selection of the feed during cutting as shown in Fig 26.

Fig 26a In the rounded edge of the tool and the workpiece surface began to contact, the cutting force increased with time, reaching a certain elastic deformation of the rake face contact as shown in Fig 26b until it reached the yield limit of plastic deformation, the formation of the maximum cutting force began to crack expansion and plastic slip under the steady state cutting as shown in Fig 26c.

From Fig 26, it can be seen that the cutting effect is not the same in different cutting stages, in the rounded edge contact stage due to only elastic deformation, the cutting can be observed in the TC4 sample slightly downward

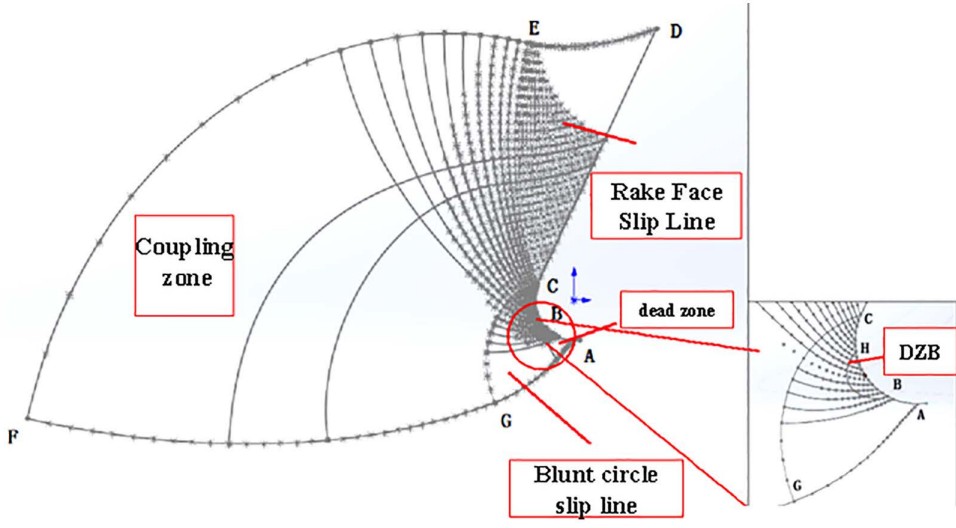

**Fig 25. Dead metal zone in the slip line field.**

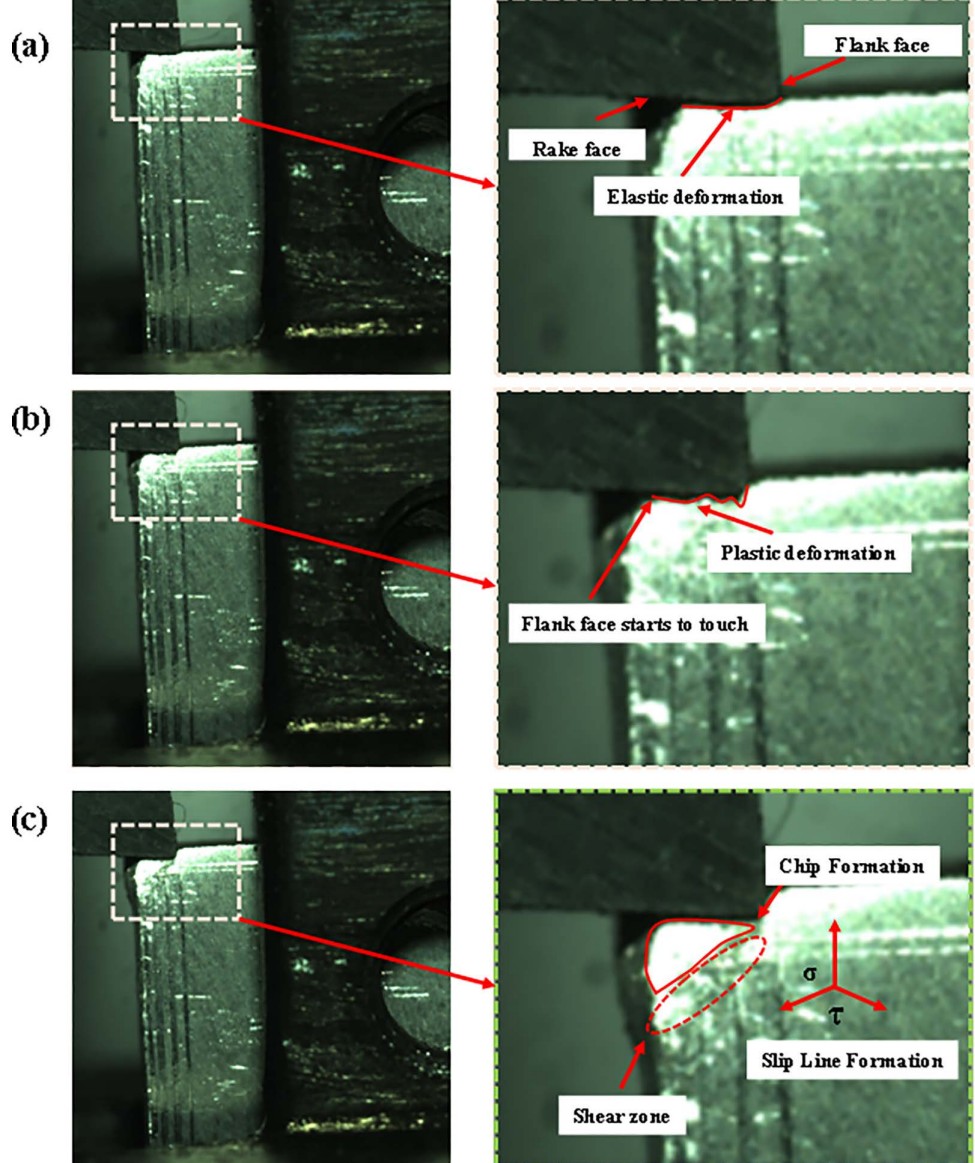

**Fig 26. Schematic diagrams for different cutting conditions: (a) rounded edge contact condition (b) rounded edge-rake face contact condition (c) steady state cutting condition.**

concave, to reach the critical moment of elastic-plastic deformation will get the maximum cutting force; when cutting into the rounded edge – rake face contact stage when the plastic deformation occurs, which is equivalent to the tool on the material to cause irreversible damage, the machined parts With the direction of the maximum shear stress will produce plastic slip, in line with the basic theory of the slip line field; into the steady state cutting, the cutting process is relatively stable, mainly manifested in the cutting force and the relative stability of the change of stress in all directions.

The slip line is solved and plotted by SolidWorks, and the slip line model is verified by selecting the stable cutting phase, by plotting the theoretical slip line and the plastic deformation during cutting as shown in Fig 27:

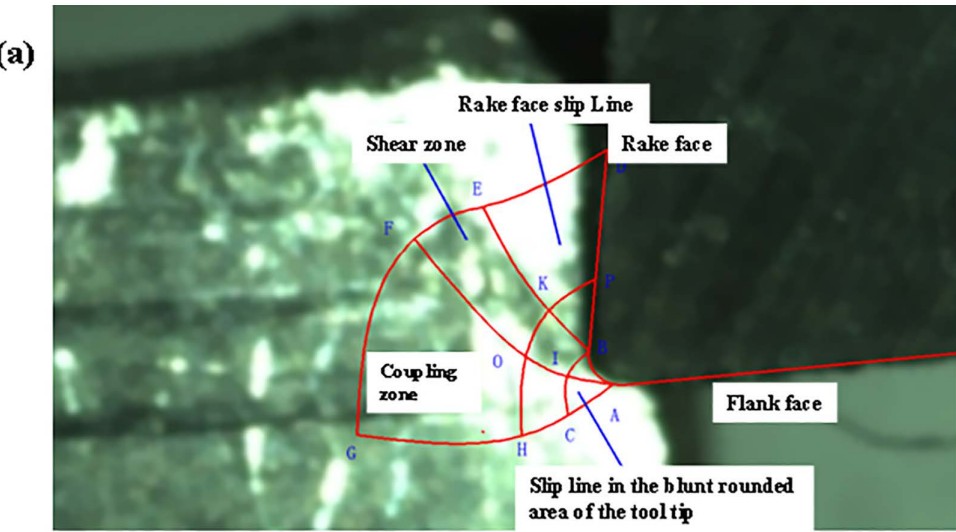

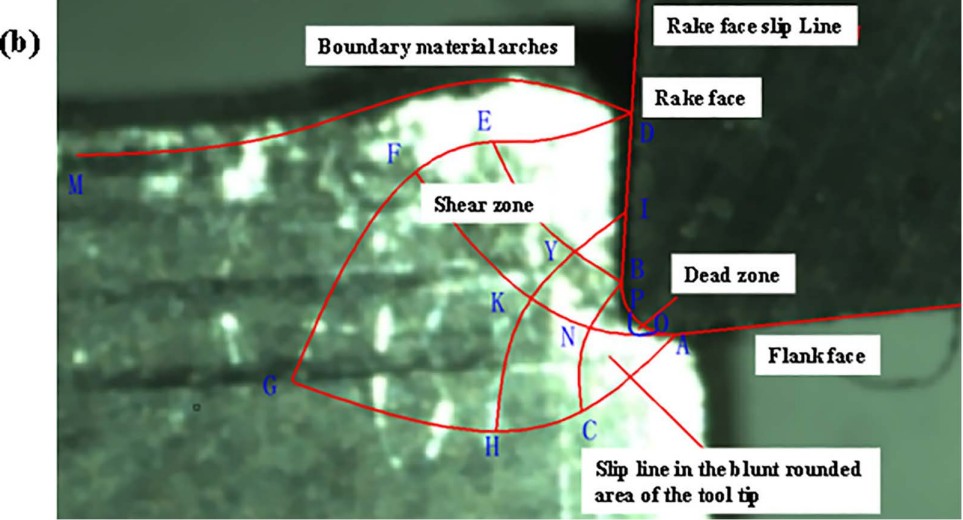

**Fig 27. Slip line field analysis for different rake angles: (a) Slip line field with 10° rake angle (b) Slip line field with 15° rake angle.**

By capturing the steady state cutting process under the high-speed camera, the corresponding slip line model is brought into Fig to fit well, and the formation of the main shear zone and serrated chips can be observed. Based on this paper completed the construction of the slip line model, based on SolidWork's secondary development of the numerical solution and the use of orthogonal cutting experiments to verify the theoretical model of the three stages, proved the correctness of the theory and the trend of the slip line in the cutting process, laying the foundation for the subsequent research Fig 28.

## 5. Conclusion

In this paper, taking TC4 as the research object and Cauchy's problem as the entry point, we analyze the research of plastic slip, dead metal zone, and stagnation point in the cutting process, and the main summarized contents are as follows:

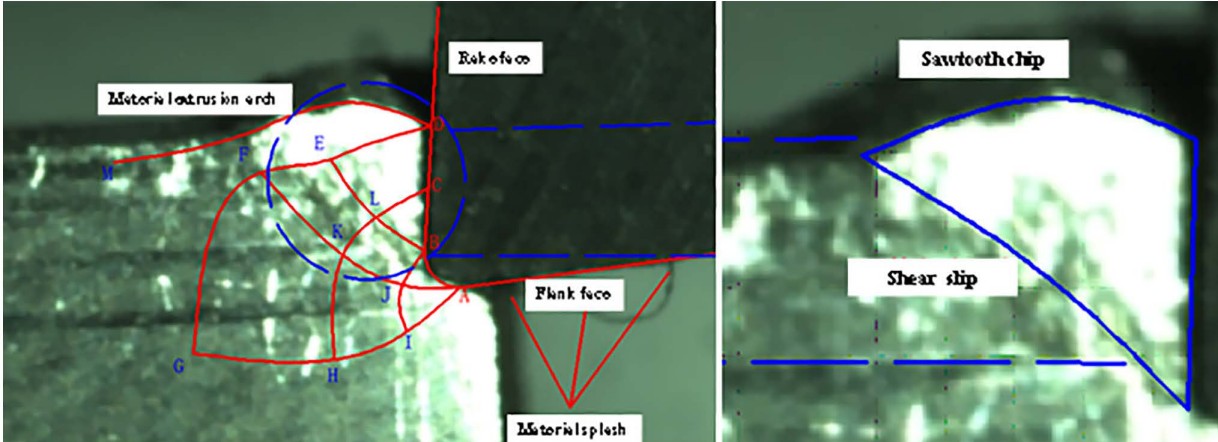

**Fig 28. Slip line field analysis during steady-state cutting.**

1. according to the mechanical analysis of the rounded edge contact in the cutting process and the mechanical analysis of the rounded edge front tool surface contact to establish the slip line field model, solve the theoretical model with the Cauchy problem, and carry out the slip line field plotting through the secondary development of the interface in SOLIDWORKS. This process completely explains the plastic slip evolution trend and the formation of the three major shear zones in the cutting process, which provides a theoretical basis for the subsequent study of chip formation, workpiece fracture, and tool life.

2. According to the stress analysis of the rounded edge contact stage of the cutting process, the critical bar of damage occurs based on the maximum elongation line strain theory the establishment of the dead metal zone and stagnation point model, and the dead metal zone model is solved by Matlab. The dead metal zone model and stagnation point model from the perspective of fracture and stress explain the dead metal zone forming boundary conditions and the shunt phenomenon of the material in the machining process, to provide a theoretical basis for the subsequent study of the built-up edges, tool wear, and surface machining quality.

## Author contributions

**Conceptualization:** Bo Hu, Sen yuan.

**Data curation:** Bo Hu, Pengfei Tian, Nian Xiao, Sen yuan, Xianfeng Zhao.

**Formal analysis:** Xianfeng Zhao.

**Investigation:** Zichuan Zou, Nian Xiao, Xianfeng Zhao.

**Methodology:** Bo Hu, Nian Xiao, Xianfeng Zhao.

**Project administration:** Bo Hu.

**Resources:** Zichuan Zou, Nian Xiao, Sen yuan.

**Software:** Zichuan Zou, Pengfei Tian.

**Supervision:** Pengfei Tian.

**Visualization:** Bo Hu.

**Writing – original draft:** Bo Hu, Xianfeng Zhao.

**Writing – review & editing:** Bo Hu, Sen yuan, Xianfeng Zhao.

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
