## [Decision Letter · Decision Letter 0]

30 Sep 2025

Dear Dr. HU,

Thank you for submitting your manuscript to PLOS ONE. After careful consideration, we feel that it has merit but does not fully meet PLOS ONE’s publication criteria as it currently stands. Therefore, we invite you to submit a revised version of the manuscript that addresses the points raised during the review process.

**ACADEMIC EDITOR: **

Please find the comments of the reviewers and please provide necessary changes for the same and submit by the date below.

We look forward to receiving your revised manuscript.

Kind regards,

Gaurav Ashok Bhaduri

Academic Editor

PLOS ONE

Journal Requirements:

“the Guizhou Provincial Youth Science and Technology Talents Growth Project(Grant No.QJJ[2024]163) ;Higher Education Engineering Research Center of Guizhou Province (Grant No. QJJ[2023]040);Guizhou Association for Science and Technology New Quality Qianyan Leading Project - Youth Voyage Program 2025 (XZQYXM-01-10).”

Please state what role the funders took in the study. If the funders had no role, please state: 'The funders had no role in study design, data collection and analysis, decision to publish, or preparation of the manuscript.'

“The authors would like to deeply appreciate the support from Higher Education Engineering Research Center of Guizhou Province (Grant No. QJJ[2023]040);the Guizhou Provincial Department of Education 2025 “Hundreds of Schools and Thousands of Enterprises Science and Technology Tackling Unveiled” Program;Guizhou Provincial Youth Science and Technology Talents Growth Project(Grant No.QJJ[2024]163) ;Guizhou Association for Science and Technology New Quality Qianyan Leading Project - Youth Voyage Program 2025 (XZQYXM-01-10”

“the Guizhou Provincial Youth Science and Technology Talents Growth Project(Grant No.QJJ[2024]163) ;Higher Education Engineering Research Center of Guizhou Province (Grant No. QJJ[2023]040);Guizhou Association for Science and Technology New Quality Qianyan Leading Project - Youth Voyage Program 2025 (XZQYXM-01-10).”

6. Please update your submission to use the PLOS LaTeX template. The template and more information on our requirements for LaTeX submissions can be found at http://journals.plos.org/plosone/s/latex .

7. We note that your Data Availability Statement is currently as follows: All relevant data are within the manuscript and its Supporting Information files

8. When completing the data availability statement of the submission form, you indicated that you will make your data available on acceptance. We strongly recommend all authors decide on a data sharing plan before acceptance, as the process can be lengthy and hold up publication timelines. Please note that, though access restrictions are acceptable now, your entire data will need to be made freely accessible if your manuscript is accepted for publication. This policy applies to all data except where public deposition would breach compliance with the protocol approved by your research ethics board. If you are unable to adhere to our open data policy, please kindly revise your statement to explain your reasoning and we will seek the editor's input on an exemption. Please be assured that, once you have provided your new statement, the assessment of your exemption will not hold up the peer review process.

9. Please ensure that you refer to Figure 2, 3, 4 and 28 in your text as, if accepted, production will need this reference to link the reader to the figure.

10. Please upload a copy of Figure 66, to which you refer in your text on page 21 in PDF submission. If the figure is no longer to be included as part of the submission please remove all reference to it within the text.

Additional Editor Comments (if provided):

Dear Authors

Please find the comments of the reviewers

Reviewers' comments:

Reviewer's Responses to Questions

**Comments to the Author**

1. Is the manuscript technically sound, and do the data support the conclusions?

Reviewer #1: Yes

Reviewer #2: Yes

Reviewer #3: Yes

Reviewer #4: Partly

2. Has the statistical analysis been performed appropriately and rigorously?

Reviewer #1: Yes

Reviewer #2: Yes

Reviewer #3: Yes

Reviewer #4: No

3. Have the authors made all data underlying the findings in their manuscript fully available?

Reviewer #1: Yes

Reviewer #2: Yes

Reviewer #3: Yes

Reviewer #4: Yes

4. Is the manuscript presented in an intelligible fashion and written in standard English?

Reviewer #1: Yes

Reviewer #2: Yes

Reviewer #3: Yes

Reviewer #4: No

Reviewer #1: 1. The paper identifies the tool rake angle (≤15°) and tool tip radius as key factors influencing the stagnation point. It is recommended to supplement a quantitative analysis of stress triaxiality and Lode angle under different cutting speeds, comparing the applicability differences of J-C, H-M, and MMC fracture criteria under dynamic loading conditions. Experimental validation can reference research on the correlation between wear mechanisms and stress distribution of high-entropy alloy-coated tools in micro-cutting TC4 titanium alloy.

2. The current model visualizes the slip-line field through secondary development in SOLIDWORKS. It is suggested to introduce a multi-scale finite element simulation approach: at the macroscopic scale, coupling the thermomechanical constitutive model for diffusion bonding of TC4 titanium alloy while considering the effect of initial welding thermal-mechanical coupling on subsequent cutting deformation; at the microscopic scale, embedding the dislocation density evolution equation to characterize the synergistic effects of α→β phase transformation and void toughening mechanisms within the adiabatic shear band. A comparative analysis of prediction error ranges between the traditional slip-line field theory and the improved model is needed.

3. Regarding the applicability threshold of the 15° rake angle in the dead-zone model, a phased validation strategy is recommended: for the low strain rate phase (<0.1 s⁻¹), reference in-situ observations of grain refinement and dynamic recrystallization behavior during TC4 hot rolling; for the high strain rate phase (>1 s⁻¹), incorporate the microvoid evolution patterns in adiabatic shear bands under one-dimensional strain compression to calibrate the correlation between slip-band inclination and temperature field distribution.

4. The paper focuses on fundamental mechanisms but should supplement an evaluation of TC4 cutting parameters' effects on the surface integrity of aerospace components. For example, comparing grain boundary migration differences between conventional rolling processes (ε = 0.0001~0.1 s⁻¹) and cutting-induced dynamic strain rates, quantifying the impact of dead-zone size on residual stress and fatigue life of the machined surface.

Reviewer #2: In general, this manuscript is suggested to be accepted for publication after minor revision.

It is recommended to take notice the following:

1. A little bit of rearranging of the abstract is needed maybe. Abstract is usually divided into four parts WHY, WHAT, HOW, and main conclusions.

WHY → This section usually contains one or two lines mainly defining what is the objective of the study or this work was done.

WHAT → This is the main portion of the abstract. It contains what was done. Like what simulations have been performed what kind of parametric studies are done to support the WHY section.

HOW → In this section you will define how u have achieved the WHAT section points. What kind of methodologies you have utilized to achieve the goals defined in WHAT section?

Usually at the end you will include one or two lines that how it is going to benefit the scientific community or what are the readership of this paper.

2. The objective of this work should be modified in the end of introduction.

3. What is the criteria for selection of parameters given in Table 1?

4. Provide the reference for the equations you used.

5. Provide the details about the FEM model developed.

6. Add more information about the results from dead metal zone morphology for different cutting parameters.

7. It is hard to follow the Slip line field for different rake angles, explain more.

8. A short subsection on “Future Research Directions” in the conclusion.

Reviewer #3: 1.The title has a weak correlation with keywords, and the keywords should be optimized.

2.The abstract only describes the model construction process but does not clarify the research objectives.

3.The paper only verifies the accuracy of the dead metal zone model when the rake angle ≤ 15°, but does not clearly explain the specific reasons for model failure when the rake angle > 15°.

4.The experiment only uses a single working condition to verify the slip-line field, without covering different rake angles.

5.Some research work can be useful in the introduction. Effect of polymeric aluminum chloride waste residue and citric acid on the properties of magnesium oxychloride cement.

6.The first point in the conclusion highly repeats the content in the middle of the abstract, both describing the model construction process.

7.The conclusion does not clearly propose process optimization suggestions for cutting TC4 titanium alloy.

Reviewer #4: The study addresses slip-line field theory in TC4 titanium alloy cutting, focusing on dead metal zone and stagnation point, but the novelty is limited since many classical models already exist.

Theoretical derivations use assumptions on stress distribution and elastic–plastic transition without rigorous justification, making the model appear weak in scientific grounding.

Simulation work through SolidWorks is constrained to narrow parameter ranges and lacks quantitative validation; advanced FEM platforms would provide more reliable results.

Differences between theoretical predictions and simulation values of the dead metal zone are significant, and the explanation for these discrepancies is not convincing.

Experimental validation is very limited, with narrow cutting conditions, and results are more illustrative than quantitative; clear visualization of the dead metal zone is missing.

No systematic error analysis or mesh sensitivity study is presented, which reduces the reliability of the simulation findings.

Literature review is descriptive rather than critical, with insufficient discussion of how the proposed model improves upon or differs from existing approaches.

Language quality and presentation issues reduce clarity; figures are not well-labeled, and overlap exists between background, results, and discussion.

Quantitative comparison among theory, simulation, and experiments is weak, leaving the conclusions insufficiently supported.

Overall contribution appears incremental, with insufficient evidence of a breakthrough in understanding slip-line field or cutting mechanisms.

**Do you want your identity to be public for this peer review?**  For information about this choice, including consent withdrawal, please see our Privacy Policy

Reviewer #1: No

Reviewer #2: No

Reviewer #3: No

Reviewer #4: **Yes: ** Rajkumar V

---

## [Author Response · Author response to Decision Letter 1]

13 Oct 2025

Original Manuscript ID: PONE-D-25-26088

Original Manuscript Title: “Study Of Shear-Plastic Slip Mechanism Based On TC4 Titanium Alloy”

To: Plos One

, Editor, Gaurav Ashok Bhaduri, PhD

Re: Response to reviewers

Dear Editor,

Thank you for allowing a revision of our manuscript, with an opportunity to address the reviewers’ comments. The submitted manuscript is the revised and resubmitted version of PONE-D-25-26088.

We are uploading (a) a cover letter for the revised manuscript, which describes how we have incorporated the reviewers' comments, (b) our point-by-point response to all the comments from the reviewers (response-to-reviewers), and (c) an updated manuscript with red highlighting indicating changes.

Dear Reviewers,

Thanks very much for your valuable comments on this manuscript. We believe that your comments strongly improved the quality of our manuscript and increased its research contribution. We have made modifications and noted changes with highlights for these comments in the corresponding locations of our revision manuscript. Thank you again for your comments. We respond to these comments in the next few pages.

Best regards,

The authors < Bo Hu, Zichuan Zou, Pengfei Tian, Nian Xiao, Sen Yuan, Xianfeng Zhao >.

Reviewer#1 Comments

Concern #1: The paper identifies the tool rake angle (≤15°) and tool tip radius as key factors influencing the stagnation point. It is recommended to supplement a quantitative analysis of stress triaxiality and Lode angle under different cutting speeds, comparing the applicability differences of J-C, H-M, and MMC fracture criteria under dynamic loading conditions. Experimental validation can reference research on the correlation between wear mechanisms and stress distribution of high-entropy alloy-coated tools in micro-cutting TC4 titanium alloy.

Author response: The stress distribution under varying cutting speeds has been comprehensively addressed in our separate publication, entitled "Mechanical analysis before the steady-state cutting of TC4 titanium alloy". Furthermore, as our research has primarily utilized the Johnson-Cook fracture criterion, we lack the foundational data and expertise to rigorously evaluate alternative models such as the Hosford-Coulomb (H-M) or the Modified Mohr-Coulomb (MMC) criteria at this time. We acknowledge this limitation in the current study.

Concern #2: The current model visualizes the slip-line field through secondary development in SOLIDWORKS. It is suggested to introduce a multi-scale finite element simulation approach: at the macroscopic scale, coupling the thermomechanical constitutive model for diffusion bonding of TC4 titanium alloy while considering the effect of initial welding thermal-mechanical coupling on subsequent cutting deformation; at the microscopic scale, embedding the dislocation density evolution equation to characterize the synergistic effects of α→β phase transformation and void toughening mechanisms within the adiabatic shear band. A comparative analysis of prediction error ranges between the traditional slip-line field theory and the improved model is needed.

Author response: We are grateful for the insightful comment raised by the reviewer. It is important to note that we have already recognized the value of this approach and have initiated a separate research project utilizing a novel molecular dynamics framework to tackle this specific challenge. Although this falls outside the immediate scope of the present manuscript, the initial results are highly encouraging, and we anticipate that it will yield a comprehensive understanding in the foreseeable future.

Concern #3: Regarding the applicability threshold of the 15° rake angle in the dead-zone model, a phased validation strategy is recommended: for the low strain rate phase (<0.1 s⁻¹), reference in-situ observations of grain refinement and dynamic recrystallization behavior during TC4 hot rolling; for the high strain rate phase (>1 s⁻¹), incorporate the microvoid evolution patterns in adiabatic shear bands under one-dimensional strain compression to calibrate the correlation between slip-band inclination and temperature field distribution.

Author response: We are grateful for the insightful comment raised by the reviewer. It is important to note that we have already recognized the value of this approach and have initiated a separate research project utilizing a novel molecular dynamics framework to tackle this specific challenge. Although this falls outside the immediate scope of the present manuscript, the initial results are highly encouraging, and we anticipate that it will yield a comprehensive understanding in the foreseeable future.

Concern #4: The paper focuses on fundamental mechanisms, but should supplement an evaluation of TC4 cutting parameters' effects on the surface integrity of aerospace components. For example, comparing grain boundary migration differences between conventional rolling processes (ε = 0.0001~0.1 s⁻¹) and cutting-induced dynamic strain rates, quantifying the impact of dead-zone size on residual stress and fatigue life of the machined surface.

Author response: We thank the reviewer for their pertinent suggestion. This very issue has been incorporated as a key objective in our recent research project. In our subsequent work, we plan to employ a micro-mechanical approach to the cutting process to systematically investigate the influence of cutting parameters on the formation mechanisms of subsurface defects. The findings will be rigorously validated using advanced characterization techniques, including Scanning Electron Microscopy (SEM) and Electron Backscatter Diffraction (EBSD).

Reviewer#2 Comments

In general, this manuscript is suggested to be accepted for publication after minor revision.It is recommended to take notice the following:

Author response: We extend our sincere gratitude for your encouraging comments on our work. We are committed to pursuing further rigorous and in-depth research in this field.

Concern #1: A little bit of rearranging of the abstract is needed maybe. Abstract is usually divided into four parts WHY, WHAT, HOW, and main conclusions.

WHY → This section usually contains one or two lines mainly defining what is the objective of the study or this work was done.

WHAT → This is the main portion of the abstract. It contains what was done. Like what simulations have been performed what kind of parametric studies are done to support the WHY section.

HOW → In this section you will define how u have achieved the WHAT section points. What kind of methodologies you have utilized to achieve the goals defined in WHAT section?

Usually at the end you will include one or two lines that how it is going to benefit the scientific community or what are the readership of this paper.

Author response: We would like to express our sincere gratitude for your review of our manuscript. In response to your comments on the abstract, we have revised it accordingly. The changes have been highlighted in red within the revised manuscript. Additionally, we have expanded the Conclusion section to include a discussion on future research prospects.

The stagnation point and dead metal zone during the cutting process directly influence chip formation and stress distribution, while the stress distribution during machining determines the direction of plastic slip in the material during cutting. This study addresses the current theoretical gap in understanding the dead metal zone and stagnation point during cutting. The cutting process of TC4 titanium alloy is analyzed by dividing it into two distinct stages: the tool-edge contact stage and the tool rake face contact stage. Based on the stress and pressure distributions in these two stages, a slip-line field model incorporating the dead metal zone is developed. The slip-line field is computed by solving the Cauchy problem and visualized via the SOLIDWORKS secondary development interface. The model for the dead metal zone and stagnation point was validated through finite element simulations under orthogonal cutting conditions. Subsequent machining experiments confirmed the slip-line field model and revealed the plastic slip behavior during TC4 cutting. Results indicate that the dead metal zone model provides higher predictive accuracy when the tool rake angle is ≤15°; the stagnation point is most significantly influenced by the tool rake angle and the cutting edge radius; and the proposed slip-line field model, which includes the dead metal zone, more accurately reflects the actual plastic slip phenomena in cutting. In conclusion, the dead metal zone model, stagnation point model, and slip-line field model collectively elucidate the cutting mechanism in the elasto-plastic stage, establishing a foundation for subsequent research on tool wear, chip formation, and machined surface quality.

Concern #2: The objective of this work should be modified in the end of introduction.

Author response: We extend our sincere gratitude for your review of our manuscript. As suggested, we have further clarified the research objectives in the final paragraph of the Introduction. The revisions have been highlighted in red in the resubmitted manuscript.

Concern #3: What is the criteria for selection of parameters given in Table 1?

Author response: We sincerely thank the reviewer for their assessment of our manuscript. Table 1 presents the parameters selected for simulating the orthogonal cutting process.

Concern #4: Provide the reference for the equations you used.

Author response: We sincerely appreciate the reviewer's valuable feedback. In response, we have thoroughly referenced all equations in the manuscript whose sources were previously unclear. These additions have been highlighted in red in the revised version.

Concern #5: Provide the details about the FEM model developed.

Author response: We sincerely thank the reviewer for this valuable feedback. In response, we have added a comprehensive elaboration of the simulation procedure following Table 1 in the manuscript. The newly added text has been highlighted in red for the reviewer's convenience.

Modeling is done by orthogonal cutting, the tool is selected from the library with its carbide tool, the size is set according to Table 1, the workpiece is selected from the library with its own TC4, the size is set to width (5mm) * height (10mm), the tool mesh is divided into 700 cells, the workpiece mesh is divided into 25 cells, the minimum number of steps is set to 10, the friction coefficient is set to 0.2, and the damage model used is the J-C model.

Concern #6: Add more information about the results from dead metal zone morphology for different cutting parameters.

Author response: We are grateful to the reviewer for this insightful observation. It is correct that our preliminary simulations encompassed a broader spectrum of rake angles (-20° to 25°). A critical comparison with our theoretical framework, however, demonstrated conclusive validation specifically for the 0° to 15° range. We made a strategic decision to concentrate the manuscript on this well-validated regime to provide a clear and robust narrative. The full dataset, including dead zone morphologies for all simulated angles, was comprehensively documented in the author's foundational Master's thesis [1], which this current work builds upon.

[1] Hu Bo. Study on Cutting Force of Titanium Alloy TC4 Based on Slip Line Field [D]. Guizhou University, 2022. DOI:10.27047/d.cnki.ggudu.2022.000441.

Concern #7:It is hard to follow the Slip line field for different rake angles, explain more.

Author response: We thank the reviewer for this comment. We acknowledge that the evolution of the slip-line field is indeed a complex process. A more detailed discussion of the underlying physical mechanisms is provided in the first author's Master's thesis [1] and in reference [29] cited in the manuscript.

[1] Hu Bo. Study on Cutting Force of Titanium Alloy TC4 Based on Slip Line Field [D]. Guizhou University, 2022. DOI:10.27047/d.cnki.ggudu.2022.000441.

Concern #8: A short subsection on “Future Research Directions” in the conclusion.

Author response:We sincerely thank the reviewer for this valuable suggestion. In response, we have added a section outlining future research prospects to the conclusion of the manuscript. These modifications have been highlighted in red in the revised version for your convenience.

Reviewer#3 Comments

Concern #1: The title has a weak correlation with keywords, and the keywords should be optimized.

Author response: We sincerely thank the reviewer for this valuable feedback. In response, we have carefully revised the keywords in the manuscript to better reflect the core themes and enhance the paper's relevance and discoverability.

Concern #2: The abstract only describes the model construction process but does not clarify the research objectives.

Author response: We sincerely thank the reviewer for this valuable suggestion. In response, we have revised the abstract to more clearly articulate the research objectives of this study. The modifications have been highlighted in red in the revised manuscript for your convenience.

Concern #3: The paper only verifies the accuracy of the dead metal zone model when the rake angle ≤ 15°, but does not clearly explain the specific reasons for model failure when the rake angle > 15°.

Author response: We are grateful to the reviewer for this insightful observation. It is correct that our preliminary simulations encompassed a broader spectrum of rake angles (-20° to 25°). A critical comparison with our theoretical framework, however, demonstrated conclusive validation specifically for the 0° to 15° range. We made a strategic decision to concentrate the manuscript on this well-validated regime to provide a clear and robust narrative. The full dataset, including dead zone morphologies for all simulated angles, was comprehensively documented in the author's foundational Master's thesis [1], which this current work builds upon.

[1] Hu Bo. Study on Cutting Force of Titanium Alloy TC4 Based on Slip Line Field [D]. Guizhou University, 2022. DOI:10.27047/d.cnki.ggudu.2022.000441.

Concern #4: The experiment only uses a single working condition to verify the slip-line field, without covering different rake angles.

Author response: We sincerely thank the reviewer for this insightful comment. We acknowledge that the current analysis is primarily based on a single-factor investigation. In our future work, we plan to incorporate multi-physics simulations that account for interacting mechanisms to enable a comprehensive multi-factor analysis, thereby strengthening the overall study.

Concern #5: Some research work can be useful in the introduction. Effect of polymeric aluminum chloride waste residue and citric acid on the properties of magnesium oxychloride cement.

Author response: We thank the reviewer for the valuable suggestion. The relevant references have been incorporated into the Introduction as recommended.

Concern #6:The first point in the conclusion highly repeats the content in the middle of the abstract, both describing the model construction process.

Author response: We sincerely thank the reviewer for this constructive feedback. We have reorganized the content to improve the logical flow and clarity of the manuscript. The revised sections have been highlighted in red for your convenience.

Concern #7:The conclusion does not clearly propose process optimization suggestions for cutting TC4 titanium alloy.

Author response: We are grateful for the valuable suggestion. In response, we have strengthened the Conclusion section through further deliberation and explicitly put forward the following recommendations for optimizing the cutting process of TC4 titanium alloy:

1. To mitigate work hardening and stress concentration, it is advisable to reduce the tool rake angle and cutting edge radius. This adjustment enhances tool sharpness, thereby improving cutting performance.

2. In future work, we plan to implement a molecular dynamics model to investigate the mechanical behavior of TC4 from a micro-mechanical perspective. This approach will allow us to analyze phenomena such as grain dislocation and sub-surface damage

---

## [Decision Letter · Decision Letter 1]

27 Nov 2025

Study Of Shear-Plastic Slip Mechanism Based On TC4 Titanium Alloy

PONE-D-25-26088R1

Dear Dr. Bo Hu,

We’re pleased to inform you that your manuscript has been judged scientifically suitable for publication and will be formally accepted for publication once it meets all outstanding technical requirements.

Kind regards,

Gaurav Ashok Bhaduri

Academic Editor

PLOS ONE

Additional Editor Comments (optional):

Accepted

Reviewers' comments:

Reviewer's Responses to Questions

**Comments to the Author**

Reviewer #1: (No Response)

Reviewer #2: All comments have been addressed

Reviewer #3: All comments have been addressed

Reviewer #4: All comments have been addressed

2. Is the manuscript technically sound, and do the data support the conclusions?

Reviewer #1: (No Response)

Reviewer #2: Yes

Reviewer #3: Yes

Reviewer #4: (No Response)

3. Has the statistical analysis been performed appropriately and rigorously?

Reviewer #1: (No Response)

Reviewer #2: Yes

Reviewer #3: Yes

Reviewer #4: (No Response)

4. Have the authors made all data underlying the findings in their manuscript fully available?

Reviewer #1: (No Response)

Reviewer #2: Yes

Reviewer #3: Yes

Reviewer #4: (No Response)

5. Is the manuscript presented in an intelligible fashion and written in standard English?

Reviewer #1: (No Response)

Reviewer #2: Yes

Reviewer #3: Yes

Reviewer #4: (No Response)

Reviewer #1: (No Response)

Reviewer #2: (No Response)

Reviewer #3: The stagnation point and dead metal zone in the cutting process directly or indirectly affect the chip formation and stress distribution, while the stress distribution in the machining process determines the plastic slip direction of the material.

Reviewer #4: (No Response)

**Do you want your identity to be public for this peer review?** For information about this choice, including consent withdrawal, please see our Privacy Policy

Reviewer #1: No

Reviewer #2: No

Reviewer #3: No

Reviewer #4: **Yes: ** Rajkumar Velu

---

## [Editor Report · Acceptance letter]

PONE-D-25-26088R1

PLOS One

Dear Dr. HU,

I'm pleased to inform you that your manuscript has been deemed suitable for publication in PLOS One. Congratulations! Your manuscript is now being handed over to our production team.

Kind regards,

on behalf of

Dr. Gaurav Ashok Bhaduri

Academic Editor

PLOS One